# Comparison of Nozzle-Based and Nozzle-Free Electrospinning for Preparation of Fast-Dissolving Nanofibers Loaded with Ciprofloxacin

**DOI:** 10.3390/pharmaceutics14081559

**Published:** 2022-07-27

**Authors:** Luca Éva Uhljar, Areen Alshweiat, Gábor Katona, Michael Chung, Norbert Radacsi, Dávid Kókai, Katalin Burián, Rita Ambrus

**Affiliations:** 1Faculty of Pharmacy, Interdisciplinary Excellence Centre, Institute of Pharmaceutical Technology and Regulatory Affairs, University of Szeged, Eötvös Street 6, 6720 Szeged, Hungary; uhljar.luca.eva@szte.hu (L.É.U.); katona.gabor@szte.hu (G.K.); 2Department of Pharmaceutics and Pharmaceutical Technology, Faculty of Pharmaceutical Sciences, The Hashemite University, Zarqa 13133, Jordan; areen.alshweiat@hu.edu.jo; 3School of Engineering, Institute for Materials and Processes, The University of Edinburgh, King’s Buildings, Edinburgh EH9 3FB, UK; michael.chung@ed.ac.uk (M.C.); n.radacsi@ed.ac.uk (N.R.); 4Department of Medical Microbiology and Immunobiology, University of Szeged, Dóm Square 10, 6720 Szeged, Hungary; kokai.david@med.u-szeged.hu (D.K.); burian.katalin@med.u-szeged.hu (K.B.)

**Keywords:** ciprofloxacin, long-term stability, polyvinylpyrrolidone, Raman mapping, nanofibers, nozzle-free electrospinning

## Abstract

The study aimed to prepare ciprofloxacin-loaded polyvinylpyrrolidone electrospun nanofibers for oral drug delivery, using a conventional nozzle-based and a lab-built nozzle-free electrospinning equipment. To produce nanofibers, electrospinning is the process most often used. However, from the industry’s point of view, conventional electrospinning does not have sufficiently high productivity. By omitting the nozzle, productivity can be increased, and so the development of nozzle-free processes is worthwhile. In this study, a solution of ciprofloxacin and polyvinylpyrrolidone was electrospun under similar conditions, using both single-nozzle and nozzle-free methods. The two electrospinning methods were compared by investigating the morphological and physicochemical properties, homogeneity, in vitro drug release, and cytotoxicity. The stability of the nanofibers was monitored from different aspects in a 26 month stability study. The results showed that the use of the nozzle-free electrospinning was preferable due to a higher throughput, improved homogeneity, and the enhanced stability of nanofiber mats, compared to the nozzle-based method. Nevertheless, fast dissolving nanofibers loaded with poorly water-soluble ciprofloxacin were produced by both electrospinning methods. The beneficial properties of these nanofibers can be exploited in innovative drug development; e.g., nanofibers can be formulated into orodispersible films or per os tablets.

## 1. Introduction

The pharmaceutical field is facing a tremendous challenge concerning the formulation of poorly water-soluble drugs into effective dosage forms of high bioavailability [1]. To overcome this obstacle, nanotechnology has been utilized as a promising strategy in the development of drug delivery systems [2]. Electrospinning (ES) was introduced as a novel method to prepare the polymeric nanofibers of poorly water-soluble drugs [3,4,5]. This method can produce nanofibers with diameters from the nano level to the submicron level, with a large surface area-to-volume ratio [6]. Moreover, the amorphous solid dispersion of drugs could be formed using an organic solution in the preparation method. Accordingly, enhanced solubility and dissolution could be achieved [7]. These properties are beneficial for orodispersed drugs. Due to their high porosity, nanofibrous oral films dissolve rapidly in saliva, disintegrate quickly, and provide rapid drug release [8,9].

The simplest method for drug loading into nanofibers is by dissolving or suspending the drug and polymer in a solvent or mixture of solvents, and electrospinning them together. The conventional single-nozzle ES setup contains a high-voltage power supply, a nozzle, a collector, and a polymer solution or melt filled in a syringe [10]. As the first step of the ES process, a droplet appears at the nozzle tip, and it is deformed into a conical structure known as a Taylor cone under the strong electric field generated by a high voltage difference. At a critical voltage, the repulsive electrostatic force overcomes the surface tension, enabling charged fluid jets to be ejected from the tip of the Taylor cone. These charged jets fly toward the grounded collector in a whipping motion, elongate, solidify, and finally deposit on the collector, resulting in fine fibers [11]. Different ES nozzle configurations have been reported based on the arrangement of the setup, such as horizontal, vertical-upward, and vertical-downward [12,13,14]. The vertical-downward configuration is used for research purposes due to its simple operation and process monitoring. The main drawbacks of the conventional setup include clogging of the nozzle and low output. Therefore, low production is the main obstacle to the use of nozzle-based ES setup for industrial applications [15].

The easiest way to increase the productivity of the ES process is to multiply the number of nozzles, which is called multi-nozzle ES [15,16]. However, this procedure also requires the use of nozzles, which tend to clog. Moreover, the unstable electric field strength and the heterogenous fiber size distribution are also disadvantages of the multi-nozzle ES configuration [15]. Based on this evidence, eliminating the use of nozzles offers a suitable alternative to overcome these issues, which led to the development of various nozzle-free ES techniques. As mentioned above, in single-nozzle ES, the Taylor cone is formed at the tip of the nozzle. In order to create a Taylor cone without a nozzle, the ES solution must be placed in a reservoir so that the Taylor cones can be generated on the surface of the solution [15,16,17,18]. To effectively create Taylor cones, a thin layer of liquid from the solution reservoir needs to be separated. This can either be achieved through methods such as bubbling [19], ultrasound [20], magnetic fields [21], corona-ES [22], or using a wire [23] or a rotating body half-immersed in the reservoir [24]. The latter concept is also the basis of the nozzle-free ES device, in which nanofibers are formed from the surface of a rotating cylinder [25,26,27].

A model drug, ciprofloxacin (CIP), a broad-spectrum fluoroquinolone antibiotic, was used in this study. From a chemical point of view, CIP is a zwitterion that possesses a base function (pKa 6.0) and an acid function (pKa 8.8) with an isoelectric point of 7.4 at 25 °C. As a zwitterion, CIP has a pH-dependent solubility with a minimum near to the isoelectric point. The solubilities of CIP in pH 6.8 and 7.4 phosphate buffer solution (PBS) were reported to be 0.17 and 0.16 mg/mL, respectively [28]. Similarly, in a previous study conducted by the research group, the solubility of CIP in pH 7.4 PBS was found to be 0.099 mg/mL [29]. Additionally, the intrinsic solubility of CIP is rated as practically insoluble, with an aqueous solubility of 0.67 mg/mL and it has a solubility in ethanol of 0.46 mg/mL at 20 °C [30]. It is well known that absorption and bioavailability are related to the solubility of poorly water-soluble drugs. Therefore, several attempts were reported to enhance the solubility of CIP, to improve its activity. Choi et al. reported an improved solubility and activity of CIP via complexation with cyclodextrin derivatives [31]. Other studies reported the fabrication of CIP through solid dispersion [32,33,34]. Moreover, nanotechnology has developed various systems for delivery of CIP, such as solid lipid nanoparticles [35] and polymeric nanoparticles [36]. The application of electrospinning to prepare CIP nanofibers was also reported in the literature; for example, Masoumi et al. [37] reported CIP/cyclodextrins based on poly (ε-caprolactone) nanofibers, which showed an enhanced CIP solubility and release compared to the pure drug. Modgil et al. [38] reported on the high permeability of CIP-loaded polyvinyl alcohol ultrafine nanofibers compared to the raw drug. Apart from this, electrospinning was employed to incorporate CIP into fibers and mats for wound dressing purposes. Kyziol et al. [39] reported biphasic kinetics for the release of CIP from alginate nanofibers loaded with CIP hydrochloride. Microbiological studies of CIP-containing nanofibers are already reported in the literature, with CIP hydrochloride/dextran/polyurethane nanofibers demonstrating good activity against Gram-positive and Gram-negative bacteria [40].

Polyvinylpyrrolidone (PVP) or povidone is one of the most commonly utilized pharmaceutical polymers. It can be found as a binder in traditional pharmaceutical products such as tablets, but it is also widely used in nanotechnology as a drug delivery polymer. PVP is in favor because it is a non-toxic, biocompatible, biodegradable polymer that is easy to handle [41]. In addition to other nanosystems, PVP is a commonly used polymer for nanofibers developed as drug carriers [42,43,44,45,46,47,48]. PVP is a water-soluble polymer, which makes it well suited for use in orodispersible or rapid release formulations.

Stability is always a key issue for drug delivery nanosystems. There are some stability studies on electrospun nanofibers in the literature, most of which focus only on the crystal structure of the drug (Table 1). Few of these are long-term stability studies, but rather, 3–6 months.

According to the authors’ best knowledge, there is no stability study of electrospun materials regarding the morphology, crystallinity, and the in vitro dissolution of the nanofibers, or a stability study investigating a time period of more than 1 year.

In the present study, nozzle-based and nozzle-free ES setups were used to prepare CIP-loaded nanofibers. The effects of the production method on the nanofiber morphology, fiber diameter distribution, structural characteristics, homogeneity, in vitro drug release, and cytotoxicity were studied and compared. Special attention has been paid to the stability of the nanofibers. The 26 month study included not only crystallographic, but also morphological and in vitro drug release experiments.

## 2. Materials and Methods

### 2.1. Materials

Ciprofloxacin powder (CIP; Mw = 331.35; purity > 98%) was donated by Teva Pharmaceutical Works Ltd. (Debrecen, Hungary) for research work. Polyvinylpyrrolidone (PVP; Mw = 1,300,000) was purchased from Alfa Aesar (Heysham, UK). The solvents for the ES solutions, the ethanol (99.99% purity), and the glacial acetic acid were obtained from Fisher Scientific (Loughborough, UK) and Sigma-Aldrich (Hamburg, Germany), respectively. As a reference for the in vitro dissolution studies, commercially available CIP-containing filmcoated per os tablets (Ciprinol^®^ 250 mg, KRKA, d. d., Novo mesto, Slovenia) were used. Acetonitrile was purchased from Molar Chemicals (Halasztelek, Hungary). The thiazolyl blue tetrazolium bromide (MTT) reagent and sodium dodecyl sulfate were obtained from Sigma-Aldrich (Hamburg, Germany). Phosphate buffer solutions (PBS; pH 2.8 and pH 7.4) were prepared in-house. All other chemicals were analytical grade, and purified water was used.

### 2.2. Electrospinning Solutions

For the electrospinning solutions, firstly, the drug and polymer solutions were prepared separately. Polymer solutions of 5 and 10 *w*/*v*% were prepared by dissolving the Mw 1,300,000 PVP powder in ethanol, while CIP was dissolved in glacial acetic acid, producing a 20 mg/mL solution. Afterward, PVP and CIP solutions were mixed in a 1:4 volume ratio and stirred for 24 h to form homogenous solutions for both types of ES. The viscosities of the ES solutions was measured using a rheometer (Haake Rheostress 1 Rheometer; Karlsruhe, Germany) with a cone-plate configuration. The diameter of the cone was 60 mm, and the angle was 1°. The temperature of the cone and plate were kept at 21.0 °C. The apparent viscosity of the ES solutions was calculated as the average value ± S.D. under the shear rate range of 31.85–100 s^−1^.

### 2.3. Electrospinning Using a Nozzle-Based Setup

A commercially available, IME electrospinning device (IME Medical Electrospinning; Waalre, the Netherlands) was used as a single-nozzle setup for nanofiber production (Figure 1A). In our previous experiment, a similar method was used to produce PVP-based nanofibers also containing CIP [29]. By that time, the maximum drug-loading was 1.5% which was increased to 5% and 10% in this work. To achieve higher drug-loading, it was necessary to change the solvent of CIP from chloroform to glacial acetic acid.

The nozzle-based ES setup was equipped with a stainless-steel 20-gauge nozzle, as the tip of a 2 mL syringe filled with the electrospinning solution and was connected to a syringe pump (IME Medical Electrospinning; Waalre, the Netherlands). The nozzle was clamped to the positive electrode of a high-voltage power supply generating +20 kV. An aluminum disk was used as a collector, and it was charged with −4 kV. The nozzle-collector distance was set at 15 cm and the flow rate was kept at 1 mL/min. The chamber of the setup was at room temperature, with a relative humidity of 31–36%.

### 2.4. Electrospinning Using a Nozzle-Free Setup

The nozzle-free ES apparatus was built in-house (Figure 1B). The design of the device is detailed in a previous paper [25]. Briefly, instead of a syringe, the ES solution was kept in a polymer solution bath in which a rotating mandrel was partially immersed. At a 15 cm distance, a rotating metal collector was placed above the rotating electrode. The applied voltages were +30 kV to the rotating electrode in the solution bath and −15 kV to the collector. The electrified jets were raised from the open liquid surface towards the rotating collector towards the rotating collector, in an upward motion. The apparatus was temperature-controlled, and the experiments were performed at 21.9 °C and 33% relative humidity.

The names and the differences of the nanofibrous samples prepared both with nozzle-based and nozzle-free ES equipment are listed in Table 2.

### 2.5. Preparation of the Physical Mixture

Physical mixtures were used for the DSC and XRPD measurements as reference samples. The mixtures contained PVP powder, and 5 *w*/*w*% and 10 *w*/*w*% CIP powder. The physical mixtures were homogenized using a shaker mixer (Turbula System Schatz; Willy A. Bachofen AG Maschinenfabrik, Basel, Switzerland) under controlled conditions (50 rpm, 10 min).

### 2.6. Characterization of the Nanofibers

#### 2.6.1. Scanning Electron Microscopy (SEM)

The morphological appearance of the electrospun fibers was investigated via scanning electron microscopy (SEM; Hitachi S4700, Hitachi Scientific Ltd., Tokyo, Japan) at 10 kV. The samples were previously coated with a thin (approximately 10 nm) film of gold–palladium using a sputter coater (Bio-Rad SC 502, VG Microtech, Uckfield, UK). The SEM images were used to measure the diameters of the nanofibers. A total of 50 nanofibers from each formulation were selected, and the mean fiber diameter was measured using ImageJ 1.44p software (Bethesda, MD, USA).

#### 2.6.2. Differential Scanning Calorimetry (DSC)

Differential scanning calorimetry (DSC; Mettler Toledo 821e DSC; Mettler Inc., Schwerzenbach, Switzerland) was applied to evaluate the thermal behavior of the samples. Approximately 3–5 mg of the nanofibrous samples were measured in the temperature interval of 30–300 °C at a heating rate of 5 °C/min, under a constant argon flow of 150 mL/min.

#### 2.6.3. Fourier-Transform Infrared Spectroscopy (FTIR)

Fourier transform infrared spectroscopy (FTIR; Thermo Nicolet AVATAR 330, Madison, WI, USA) was performed after discs of a KBr and nanofibers had been made. The samples were ground with 150 mg dry KBr in a mortar and pestle, then compressed into a disc with 10 t pressure. The discs were scanned 128 times over the range of 4000–400 cm^−1^ and with a resolution of 4 cm^−1^.

#### 2.6.4. X-ray Powder Diffraction (XRPD)

The crystalline s of the nanofibers, the physical mixtures, and the raw drug were characterized using an X-ray powder diffraction system (XRPD; BRUKER D8 Advance diffractometer, Karlsruhe, Germany). The samples were measured with Cu Kα radiation (λ = 1.5406 Å), scanned at 40 kV and 40 mA in the interval of 3–40 2θ, and a VÅNTEC-1 detector was used. All manipulations (Kα2-stripping, background removal, and smoothing) were performed with DIFFRAC plus EVA software (Karlsruhe, Germany).

#### 2.6.5. Homogeneity

To obtain information regarding the homogeneity of the nanofibrous mats, specimens of the mats (0.5 × 0.5 cm^2^) were cut from the edge and the center of the collector. Raman spectra of the solid specimens were recorded with a Thermo Fisher DXR dispersive Raman microscope (Waltham, MA, USA) equipped with a CCD camera and a diode laser operating at a wavelength of 780 nm. The applied laser power was 24 mW at a slit aperture size of 50 µm. The spectra were collected from a 400 × 400 µm^2^ area with a 50 µm step size in the spectral range of 200–3300 cm^−1^. The exposure time was 6 s and an average of 8 scans was applied to construct Raman chemical map. Determining the distribution of CIP in the specimens, the spectral range from 1650 to 1550 cm^−1^ was selected for profiling (Appendix A). The data were collected using automated fluorescence corrections. For the removal of cosmic rays, a convolution filter was applied to the original spectrum using Gaussian kernel. OMNIC 8 software was used for the data collection.

In addition to the Raman measurements, the drug loading (DL) and entrapment efficiency (EE) of the specimens collected from the edge and the center of the collector were also investigated. The DL and EE were quantified spectrophotometrically (ABL&E-Jasco UV/VIS Spectrophotometer V-730, Budapest, Hungary) at 271 nm after dissolving the specimens in pH 7.4 PBS.

DL and EE of the nanofibrous formulations were calculated according to the following equations:(1)$$\mathrm{DL}\;\left(\%\right)=\frac{\mathrm{measured}\;\mathrm{mass}\;\mathrm{of}\;\mathrm{CIP}\;\left(\mu g\right)}{\mathrm{nanofiber}\;\mathrm{weight}\;\left(\mu g\right)}\times 100$$

The nanofiber weight was obtained by weighing the 0.5 × 0.5 cm^2^ specimens. After the dissolution of the specimens in pH 7.4 PBS, the measured mass of CIP was calculated from the UV absorption of the drug.
(2)$$\mathrm{EE}\;\left(\%\right)=\frac{\mathrm{measured}\;\mathrm{CIP}\;\mathrm{concentration}\;\left(\frac{\mu g}{\mathrm{mL}}\right)\;}{\mathrm{total}\;\mathrm{CIP}\;\mathrm{concentration}\;\left(\frac{\mu g}{\mathrm{mL}}\right)}\times 100$$

The measured CIP concentration was calculated from the UV absorption of CIP after weighing and dissolving the specimens in pH 7.4 PBS. In addition, the initial CIP concentration in the electrospinning solution was considered as the total CIP content.

Each experiment was performed in three parallel measurements, and the average values and standard deviations are reported.

#### 2.6.6. In Vitro Drug Release Studies

As part of the comparison of the nozzle-based and the nozzle-free ES, in vitro drug-release studies were carried out. A modified paddle method (Hanson Research SR8-Plus release device; Hanson Research, Chatsworth, CA, USA) was used to measure the drug release from the different nanofibers (containing 25 mg CIP) compared with 25 mg raw CIP powder, and Ciprinol^®^ 250 mg tablets. The release studies were carried out in 50 mL of pH 7.4 PBS medium at 37 °C. The paddle was rotated at 100 rpm. Samples of 0.5 mL volume were taken manually from the buffer solution after 1, 3, 5, 10, 15, 20, 30, 40, 50, 60, and 90 min. After sampling, the volume was replaced with fresh PBS.

The concentration of the drug present in the aliquots was determined with high performance liquid chromatography (HPLC; Agilent 1260 HPLC, Agilent Technologies, Santa Clara, CA, USA). Chromatographic separation was performed using a Kromasil^®^ C18 (4.6 × 250 mm, 5 µm; Nouryon, Bohus, Sweden) analytical column at 25 °C. The mobile phase consisted of (A) an aqueous solution of 0.02 M KH_2_PO_4_ adjusted to pH = 2.8 with ortho-phosphoric acid and (B) acetonitrile with isocratic elution (A:B; 77:23; *v*/*v*) at a flow rate of 1 mL/min. The analytical time was 5 min for each injection. A sample volume of 20 μL was injected to determine the concentration of CIP at 210 nm, using a UV/VIS diode array detector. Data were evaluated using ChemStation B.04.03. Software (Agilent Technologies, Santa Clara, CA, USA). The retention time of the drug was 3.57 min (Appendix A). The concentration of CIP was calculated with the help of a calibration curve in the concentration range of 1–1000 μg/mL (Appendix A). The regression coefficient (R^2^) of the calibration curve was 0.998.

#### 2.6.7. Study of Drug-Release Kinetics and Mechanism

The release kinetics of CIP from the nanofibers were determined. The fitting of the results was tested with zero order, first order, Higuchi, the Korsmeyer–Peppas model, and the Hixson–Crowell cube root law [61]. The model for the fitting with the largest R^2^ was accepted as the release kinetics.

#### 2.6.8. Long-Term Storage Stability Tests

The nanofibrous samples were stored in a desiccator, protected from light, at room temperature for 26 months. Stability was tested after 3, 5, 8, 16, and 26 months of storage. In order to obtain external and internal information regarding the CIP-loaded nanofibers, their macrostructure was investigated using SEM images and ImageJ software, and their microstructure via XRPD. Then, in vitro drug-release studies were performed on the samples considered critical, i.e., at 8 and 16 months. The methods for the measurements were described above.

#### 2.6.9. In Vitro Cytotoxicity

Mitochondrial activity quantification as a measure of cell viability was performed via MTT assay in 96-well cell culture microplates using Caco-2 (human colorectal adenocarcinoma) cells. Caco-2 cells were seeded at a density of 4 × 10^4^ cells/well. Firstly, serial dilution was made of the following stock concentrations: 325 µg/mL ciprofloxacin solution, 1570 µg/mL SN-5 solution, 1570 µg/mL SN-10 solution, or 1520 µg/mL NF-5 solution, and then the cells were incubated with the mentioned materials at 37 °C for 24 h. Later, 20 μL of MTT reagent was added to each well. After an additional incubation at 37 °C for 4 h, 100 µL sodium dodecyl sulfate solution (10% in 0.01 M HCI) was added, and the cells were incubated overnight. The cytotoxicity of the compounds was then determined with an EZ READ 400 ELISA reader (Biochrom, Cambridge, United Kingdom) by measuring the OD at 550 nm (ref. 630 nm). The assay was repeated four times for each concentration.

#### 2.6.10. Statistical Analysis

Statistical analysis was carried out to determine the existence of a significant difference between the measured data. A one-way analysis of variance (ANOVA) with post hoc Tukey HSD test was performed on the fiber diameter and the in vitro drug-release data. For the latter, all measured points of the nanofibers, the untreated CIP powder, and the Ciprinol^®^ 250 mg tablet were compared one by one. The results with *p* < 0.01 were assumed to be statistically significant.

## 3. Results and Discussion

### 3.1. Morphology of CIP-Loaded Nanofibers Made Using a Nozzle-Based ES Apparatus

The morphology and fiber diameter distribution of the nanofibrous mats were important for the comparison of the conventional nozzle-based ES apparatus and the lab-built nozzle-free ES equipment. During single-nozzle ES, continuous fibers with smooth surfaces were formed, as shown in Figure 2. On the other hand, the SN-5 formulation had a few beads, while SN-10 formulation had many. In the latter case, large sack-shaped formations were also observed, indicating the imperfection of the ES. This may have been caused by the increase in the concentration of CIP, as nanofibers prepared in a similar way were previously found to have the appropriate morphology [29].

The average fiber diameter of SN-10 was significantly higher (*p* < 0.01) compared with SN-5 (Table 3). This more than two-fold increase in the average fiber diameter may be related to the increased viscosity of the ES solutions (SN-5_diam_. = 323 ± 51 nm, SN-10_diam_. = 735 ± 91 nm, and SN-5_visc_. = 79 ± 2 mPas, and SN-10_visc_. = 336 ± 19 mPas). The size distribution of the nanofibers was homogeneous, as the standard deviations were small, and the distribution diagrams were monodisperse and bell shaped (Figure 2). All the morphological observations and fiber diameter data are summarized in Table 3.

### 3.2. Morphology of CIP-Loaded Nanofibers Made Using a Nozzle-Free ES Apparatus

Nanofiber production from the ES solutions using the nozzle-free ES equipment was also planned. However, the NF-10 solution was too viscous, and it was not possible to produce proper nanofibers, even by varying the collector-bath distance over a range of 10–20 cm, and by varying the applied voltage between +20–30 and −15–30 kV on the rotating mandrel and the collector, respectively. The fibers stuck together and fused, resulting an incorrect product, as Figure 3 shows. Therefore, fiber diameter measurement and further tests with NF-10 sample were disregarded.

In contrast, discrete, continuous, smooth-surfaced nanofibers were produced in the case of the NF-5 formulation (Figure 3). However, the diameters of the fibers varied widely, with the SEM images showing a wide range of nanofiber thicknesses. The standard deviation of the fiber diameter and the distribution histogram both confirmed polydisperse distribution, with a large coefficient of variation (36%). The average fiber diameter was 1167 nm, which was 3.6 times larger than that of the SN-5 formulation (Table 3). However, the arithmetic mean may be affected by outliers, so it is also advisable to take the fiber diameter distribution into consideration. The histogram shows that 42% of the measured nanofibers were between 600 and 1000 nm, and 34% were between 700 and 900 nm (Figure 3). Therefore, the distribution is strongly skewed towards smaller fiber diameters. Overall, the production of nanofibers loaded with 5% CIP was achieved using nozzle-free ES equipment. The morphology of the fibers was satisfactory; however, the polydisperse distribution may require further development.

### 3.3. Structural Characterization

It is known that during the ES process the crystal structure of the active substance may change, in most cases amorphizing. The amorphizing effects of ES has been confirmed in several previous studies [29,44,49,54,55,60,62]. Therefore, nanofibers can be considered as amorphous solid dispersions. Two types of experiments, DSC and XRPD, were performed to prove that both ES methods could produce amorphous solid dispersions. Figure 4A presents the DSC thermograms of CIP powder, physical mixtures (PM-5 and PM-10), and samples made by nozzle-based ES (SN-5 and SN-10) and nozzle-free ES (NF-5). The sharp endothermic peak at 275 °C for CIP powder and physical mixtures represents the melting point of crystalline CIP. As the size of the peak depends on the amount of the crystalline material, the peak of the 5% CIP-loaded sample (PM-5) was approximately half the intensity of the peak of the 10% CIP-containing sample (PM-10). However, the peak did not appear in the thermograms of nanofibers because of the absence of a measurable melting point. Thus, all nanofibrous samples contained amorphized CIP regardless of the preparation method, and all of the produced nanofibers can be considered as amorphous solid dispersions.

In addition to the DSC measurements, the change in the crystallinity of CIP was studied using XRPD (Figure 4B). The CIP powder showed high crystallinity, as long, sharp peaks appeared on its diffractogram at around 2-Theta = 5.8, 9.1, 14.5, and 23.2°. In the case of PM-10, there were characteristic CIP peaks and a wide peak typical of PVP between 7.5 and 15.2°. In contrast, the diffractograms of the nanofibrous samples (SN-5, SN-10, and NF-5) indicated amorphous API. Hence, the sample PM-10 contained crystalline API as proof that the amorphization had occurred during production, and not because of any polymer–drug interactions.

The main interactions between the PVP and the CIP were observed via FTIR spectroscopy. Figure 4C shows a comparison of the spectra of SN-5, SN-10, NF-5, CIP, and PVP. According to the analysis, all the spectra of nanofibrous samples (SN-5, SN-10, and NF-5) had a similar shape. The characteristic peaks of CIP (1750–400 cm^−1^) appeared on the spectra, as well as the broad peaks of PVP (3700–2800 cm^−1^). The shifts and widenings confirmed the successful incorporation of the CIP into the nanofibers in all cases, regardless of the ES method.

### 3.4. Homogeneity

The homogeneity of the nanofiber mats deposited on the collector during the ES was investigated using two different methods. Both methods measured the CIP content in the specimens taken from the center and the edge of the mats.

Firstly, Raman chemical mapping was executed on the specimens (Figure 5). The warmer colors on the maps indicate a locally higher drug content. The samples made via the nozzle-free method (NF-5 center and NF-5 edge) both contained high levels of CIP, as the maps show yellow, orange, and red colors. So, not only the two maps themselves, but the whole nanofiber mat can be considered homogenous. On the other hand, the center specimens of the nozzle-based methods (SN-10 center and SN-5 center) and the SN-5 edge had medium–high CIP contents and showed noticeable variations within each map. It can be seen that green and even blue patches appeared, indicating lower CIP levels. Moreover, the drug content of the SN-10 edge specimen was poor, as the map had some yellow, but more blue spots on a green background. So, while it is likely for SN-5, for SN-10 it is clear that nozzle-based ES production resulted in less API at the edge of the nanofiber mat and more in the center. The whole mat was not homogeneous. This should be taken into consideration for future experiments and further processing, since only by using the whole fiber mat can the API content be estimated with a good approximation. If only a part of the mat is used, there may be a large error between the theoretical and the real drug content.

After the Raman mapping, the exact CIP contents of the specimens were measured via UV/VIS spectroscopy. Drug loading (DL) and entrapment efficiency (EE) were calculated from the absorbance values, using a calibration curve (R^2^ = 0.9990) (Table 4). The results confirmed the findings of the Raman mapping. In the case of the SN-10 specimens, the differences between the DL of the center and the edge were quite big, as well as the standard deviations. EE was considered to be moderately weak, and the 17% difference between the two parts of the mats was striking. The standard deviations of SN-15 center were also high. In this case, there was also a notable difference (~7%) between the EE of the center and the edge specimens. The NF-5 sample showed greater homogeneity and good EE. Here, the difference between the EE values was 4.5%, and it was accompanied by high values, namely 88.7 ± 14.2% and 93.3 ± 15.0%.

In summary, nozzle-free production results in a more homogeneous nanofiber mat gathering on the collector. The reason for this may be that the point of origin of the jets is not concentrated at a single point, as in nozzle-based ES, but ejected from several locations of the bath simultaneously.

### 3.5. In Vitro Drug-Release Studies

In vitro releases of CIP from the different nanofibers were compared with the dissolution of untreated drug powder and Ciprinol^®^ 250 mg tablet in pH 7.4 PBS. The dissolution curves are shown in Figure 6. The untreated CIP powder dissolved the least, as even after 90 min only 33% of the measured amount was dissolved. Formulating the drug into tablets resulted in a slightly better dissolution of around 40%. It is worth noting that the dissolution from the tablet was fast, and reached its maximum in 3 min. The dissolution was also fast (5–10 min) from every nanofibrous sample, and more CIP was released in 3 min than from the tablet. Additionally, the nanofibrous formulations had the advantage of facilitating 100% release of the incorporated drug within a short time. Since the PVP is a water-soluble polymer and the CIP was in its amorphous form, rapid and complete release was expected. The results were consistent with previous results using 1 *w*/*w*% CIP-loaded nanofibers [29].

A statistical analysis of the percentages measured at each time point was performed, to allow a more accurate comparison of the release (Figure 7). There was no significant difference between the dissolution of SN-5, SN-10, and NF-5 nanofibrous samples, except for the first minute. However, the released CIP percentage from the nanofibrous samples were significantly higher (*p* < 0.01) than the values of the CIP powder or the tablet at almost every sampling point. In the first few minutes of release, it was variable as to whether the difference was significant, but after 5 min, the significance was obvious. Therefore, it can be concluded that a fast and complete release of CIP from all nanofibers was achieved, and that the dissolved percentages of the CIP was significantly higher (*p* < 0.01) than in the case of the CIP powder or tablet.

### 3.6. Study of Drug-Release Kinetics

The kinetics of the in vitro drug release were determined. The regression coefficient of the models fit into the release curves can be seen in Table 5 The drug-release kinetics from nanofibers containing 5 *w*/*w*% CIP (samples SN-5 and NF-5) could be best described using the Korsmeyer–Peppas model and first-order kinetics. This strong fit to the Korsmeyer–Peppas model was also true for the drug release from the 1 w/w% CIP-loaded nanofibers studied previously [29]. Additionally, the release from the sample SN-10 was best described using the Hixon–Crowell cube root law (R^2^ = 0.9876). The release curves for raw CIP powder and commercially available tablets were best fit using the Higuchi model, although with smaller R^2^ values of 0.9148 and 0.8243, respectively. Similar to these results, it was observed in our previous experiments that the R^2^ for the raw CIP was approximately 0.9 for the Higuchi and Korsmeyer–Peppas models, while the same models were fit with a higher R^2^ value for the nanofibrous formulations [29]. The results of this study presented the same observation. In the case of the tablet, the ascending slope of the curve was very steep, since the CIP release from the tablet reached its maximum in 3 min, and this made the model fitting difficult. This may be the reason for why none of the R^2^ values were adequately high.

### 3.7. Long-Term Storage Stability

As the first step of the stability study, electron microscopy images of the nanofibrous samples stored in the desiccator were taken. Figure 8 shows the changes in the morphology of the nanofibers over a period of 26 months. For SN-10, the structural aging was estimated to be between 8 and 16 months. At 16 months, the fibers had a rough surface and were not able to retain their individuality, but merged at several points, forming a web-like structure. After 26 months, the fibrous structural elements were barely present in the sample, which had fused into a film-like structure. The fusion was confirmed by the increase in average fiber diameter (Table 6). SN-5 was also produced with single-nozzle ES, and in this case, no major morphological changes occurred, although a few merges were detected at 8 months. An increase was observed for the standard deviation of fiber diameters, which could indicate the fusion of some nanofibers. Regarding NF-5, even a large variation in fiber diameter already existed in the fresh sample, which almost doubled between 16 and 26 months. The reason for the extreme increase is that the fibers, such as NF-10, lost their individuality and formed a mesh. In summary, the morphological stabilities were 8 months, 16 months, and more than 26 months in the case of SN-10, NF-5, and SN-5, respectively.

As the second step of the stability study, the crystallinity of the CIP was monitored via XRPD measurements. As reference, CIP was dissolved in ethanol and acetic acid, and after crystallization, the powders were used. The characteristic peak of CIP-acetate at around 6 2-Theta was detected in several SN-10 nanofibrous samples, indicating that the drug had partially re-crystallized (Figure 9). At 5 months, a small acetate peak was observed, while it was dominant in the XRPD pattern at 16 and 26 months. Other characteristic peaks of CIP (14.5, 17.5, 20.5, 25.5, and 27 2-Theta) appeared for the nanofibers produced using nozzle-based ES at 8 and 16 months. According to other studies, the amorphous form of API was preserved in nanofibers over 12 months [58,60]. Based on this, the re-crystallization might have occurred between the 12th and 16th month. On the other hand, no sharp peak appeared in the case of NF-5, which indicated the amorphous character of the drug. The flat, broad peaks were signals of the polymer. Overall, based on X-ray measurements, nozzle-free ES provides better stability since CIP was amorphous until the end of the stability study, while re-crystallization took place for the SN-10 and SN-5 formulations up to the 5th and 8th month, respectively.

The morphology of the fibers can affect the processability and the dissolution of the API. The latter is possibly affected by the crystal structure of the API, as well. For this reason, in vitro drug release was investigated as part of the stability test. In the literature, there are examples for both maintained and changed drug release from nanofibers after 3 months of storage [54,55,60]. In this case as previously concluded, storage intervals of 8 and 16 months were found to be critical, so the release of the CIP was measured and compared with the fresh nanofibers. The results are shown in Figure 10. Despite SN-10 displaying stability problems based on electron microscopy and X-ray measurements, the drug release was not affected, as the total amount of CIP was dissolved in a short time. On the other hand, in the case of SN-5, the release was decreased at 16 months. It took approximately three times longer to reach 90% than for the fresh or the 8-month sample. The slower release may be explained by the formation of additional H-bonds between the API and the polymer molecules [60]. Interestingly, the re-crystallization of CIP was already visible in the XRPD pattern at 8 months (Figure 8), but the effect on dissolution was only noticeable at 16 months. A similar observation was made for nanofibers containing itraconazole, where the release did not change during the 3 months of storage, even though a phase separation of the matrix and the API has occurred [55]. In the case of NF-5, the CIP remained amorphous after 26 months, and consequently, there was no change in the in vitro dissolution. To the authors’ best knowledge, such long stability data for drug carrier nanofibers are not yet available in the literature.

Overall, in terms of drug release, the nanofibers prepared with nozzle-free ES were found to be more stable, but the limiting factor that the samples were stored in a desiccator should be taken into consideration.

### 3.8. In Vitro Cytotoxicity

The in vitro cytotoxicity was tested using an MTT assay (Figure 11). CaCo-2 cells were treated for 24 h with serial dilutions of CIP solution (325 µg/mL), SN-10 solution (1570 µg/mL), SN-5 solution (1570 µg/mL), and NF-5 solution (1520 µg/mL). It is important to note that Figure 9 shows the concentrations of the drug-loaded nanofibrous samples. The highest CIP concentrations were the following: 78,5 µg/mL in the case of SN-5, 157 µg/mL in the case of SN-10, and 76 µg/mL in the case of NF-5. The solution of the untreated CIP powder was found to be cytotoxic in 163 µg/mL concentrations, but the solutions of different nanofibers were not cytotoxic in any measured concentrations. This suggests that the CIP-loaded PVP nanofibers may be suitable for per os administration.

## 4. Conclusions

Nanofibers as amorphous solid dispersions can be formulated as orodispersed films or oral medicines, while with the suitable polymer, fast disintegration and rapid release can be achieved. This paper presented the production and investigation of CIP-loaded PVP nanofibers prepared via the nozzle-based and nozzle-free ES methods. Nanofibers with 5% and 10% CIP concentrations were fabricated using conventional single-nozzle ES. Using the nozzle-free method, 5% CIP-loaded nanofibrous samples were produced from the same electrospinning solutions. Comparing the nanofibers, we found that the preparation method had no influence on the drug carriers as amorphous solid dispersions, because both methods could amorphize the CIP. Additionally, the presence or absence of the nozzle had no effect on the in vitro drug release, as the dissolution of the CIP from the nanofibers was complete and fast, and occurred according to the Korsmeyer–Peppas model or first-order kinetics. This result was similar to the preliminary study in which CIP was present in lower concentrations in the PVP nanofibers [29]. It also correlated well with the results of CIP-loaded PVP nanofibers investigated by another research group [63]. Finally, none of the prepared nanofibrous samples were cytotoxic according to the MTT test on the CaCo-2 cell lines. This study may suggest further processing steps, which might include the production of an orodispersed film via the pressing or the production of per os drugs, by filling the fibers into gelatin capsules [64,65].

However, it can be concluded that nozzle-free ES has several advantages. These are, based on the studies performed, a more homogeneous distribution of the active ingredient within the nanofiber mat, a higher encapsulation efficiency, and a longer stability. An additional advantage may be increased productivity, due to the fact that ES from the free solution surface allows the formation of several Taylor cones at the same time. On the other hand, this also leads to the disadvantage of the nozzle-free method, which is a wide variability in fiber diameters.

The study also focused on the stability of the nanofibers. In this context, SEM, XRPD, and in vitro release studies were carried out, which led to two types of conclusions. The first conclusion was that aged morphology, the fusion of the fibers or partial re-crystallization of the CIP did not affect the in vitro drug release. Based on these results, it is important to check for drug release in all stability studies of the nanofibers. As a second conclusion, the stability results suggested that NF-5 was sufficiently stable for 16 months in terms of morphology, and for 26 months in terms of amorphousness and in vitro release. In contrast, the morphology of the SN-5 sample remained unchanged up to 26 months, but the CIP started to re-crystallize at 8 months, and its release slowed down significantly at 16 months. Thus, the nanofibers prepared with nozzle-free ES were found to be more stable during the 26 months of storage in a desiccator.

Overall, nanofiber production using nozzle-free ES equipment can not only provide a good alternative to the nozzle-based ES, but it also produces nanofibers with improved properties. Thus, it could be worthwhile to develop the method further and to use the resulting nanofibers as drug delivery systems.

## Figures and Tables

**Figure 1 pharmaceutics-14-01559-f001:**
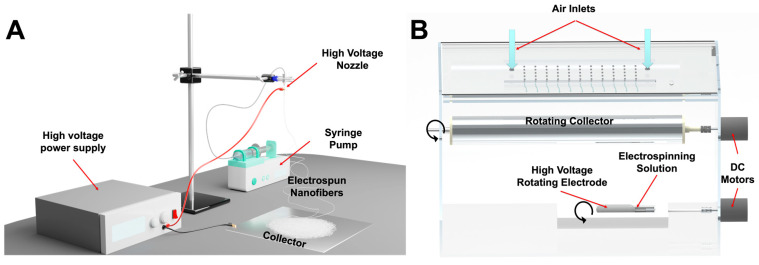
Schematic image of the two different electrospinning systems used in this study. (**A**) Conventional vertical nozzle-based electrospinning setup. A syringe pump delivers the electrospinning solution to the high-voltage nozzle. The collector plate has a negative bias. (**B**) Home-built nozzle-free high-throughput electrospinning setup. Positive high voltage is applied to the rotating electrode inside a solution bath. The rotating collector is also on high voltage, but with negative polarity. Air with controlled humidity enters the chamber from the top.

**Figure 2 pharmaceutics-14-01559-f002:**
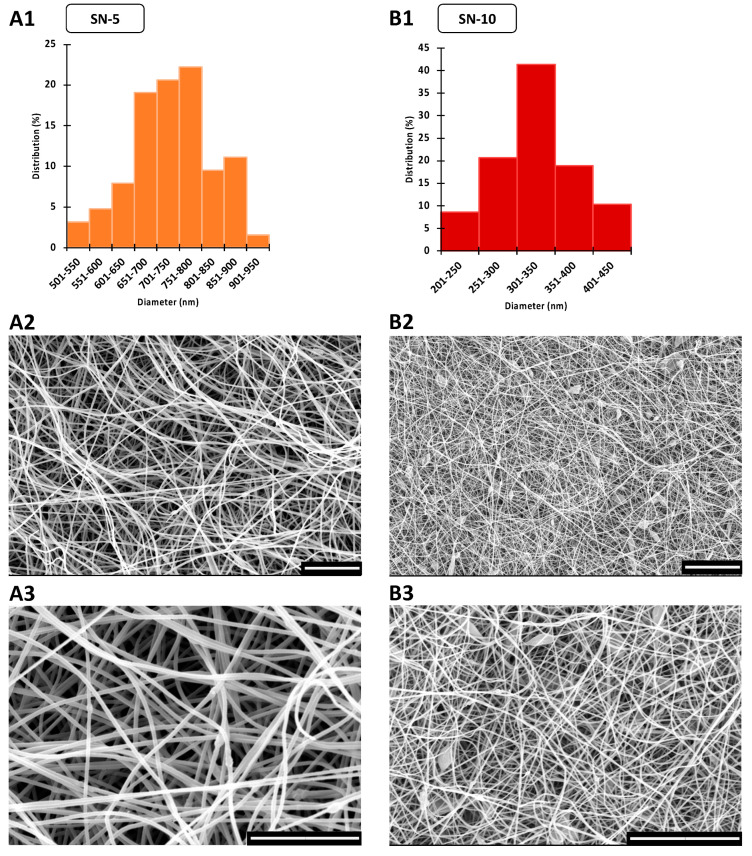
SEM images and diameter distributions of nanofibrous samples produced via nozzle-based electrospinning. The scale bar is 20 µm on every microscopic image. (**A1**) The average fiber diameter was 323 ± 51 nm in the case of nanofibers containing 5 w% ciprofloxacin (SN-5). The size distribution of nanofibers was monodisperse. (**A2**,**A3**) The morphology of the fibers was mainly uniform, but some beads formed. (**B1**) In the case of the nanofibers containing 10 w% ciprofloxacin (SN-10), the average fiber diameter was significantly larger (735 ± 91 nm). The size distribution had a bell shape. (**B2**,**B3**) In addition to the numerous beads, large sack-shaped formations also appeared proving the imperfect electrospinning process.

**Figure 3 pharmaceutics-14-01559-f003:**
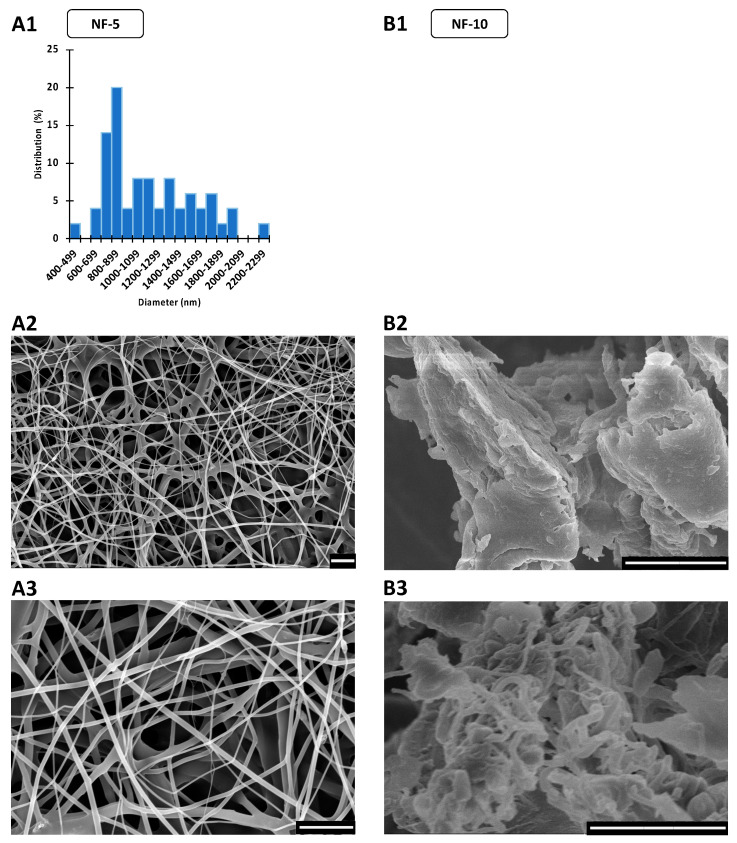
SEM images and diameter distributions of nanofibrous samples produced by nozzle-free electrospinning. The scale bar is 10 µm on every microscopic image. (**A1**) In the case of the nanofibers containing 5 w% ciprofloxacin (NF-5), the free-surface electrospinning was successful. The average fiber diameter was 1167 ± 415 nm, but the size distribution was polydisperse and strongly skewed left. (**A2**,**A3**) The fibers were mainly individual and had smooth surfaces. (**B2**,**B3**) The nozzle-free electrospinning of 10 w% ciprofloxacin containing nanofibers (NF-10) was impractical; instead of separated fibers, merged fiber balls and polymer films were prepared. (**B1**) Thus, the fiber size was not measurable and the distribution was not interpretable.

**Figure 4 pharmaceutics-14-01559-f004:**
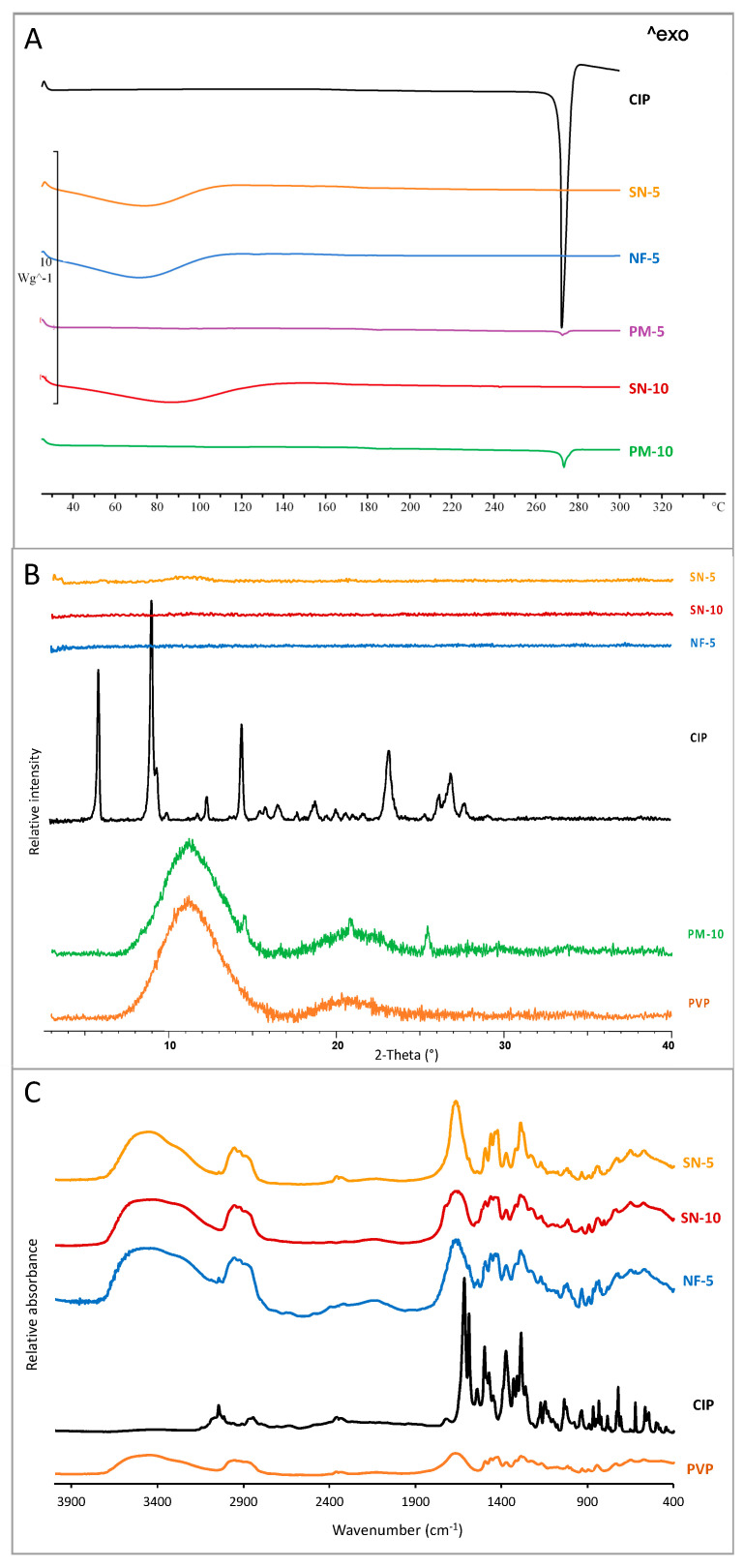
Physicochemical properties of ciprofloxacin-loaded PVP nanofibers. DSC thermograms (**A**) and XRPD diffractograms (**B**) proved that the amorphous state of the drug in the nanofibers produced both single-nozzle (SN-5 and SN-10) and nozzle-free (NF-5) electrospinning. As references, pure ciprofloxacin powder (CIP) and physical mixture (PM) were used. The successful incorporation of the drug was confirmed with FTIR spectra (**C**). The characteristic shifts and widenings showed up at the same wavenumbers, demonstrating successful incorporation in all cases, regardless of the ES method.

**Figure 5 pharmaceutics-14-01559-f005:**
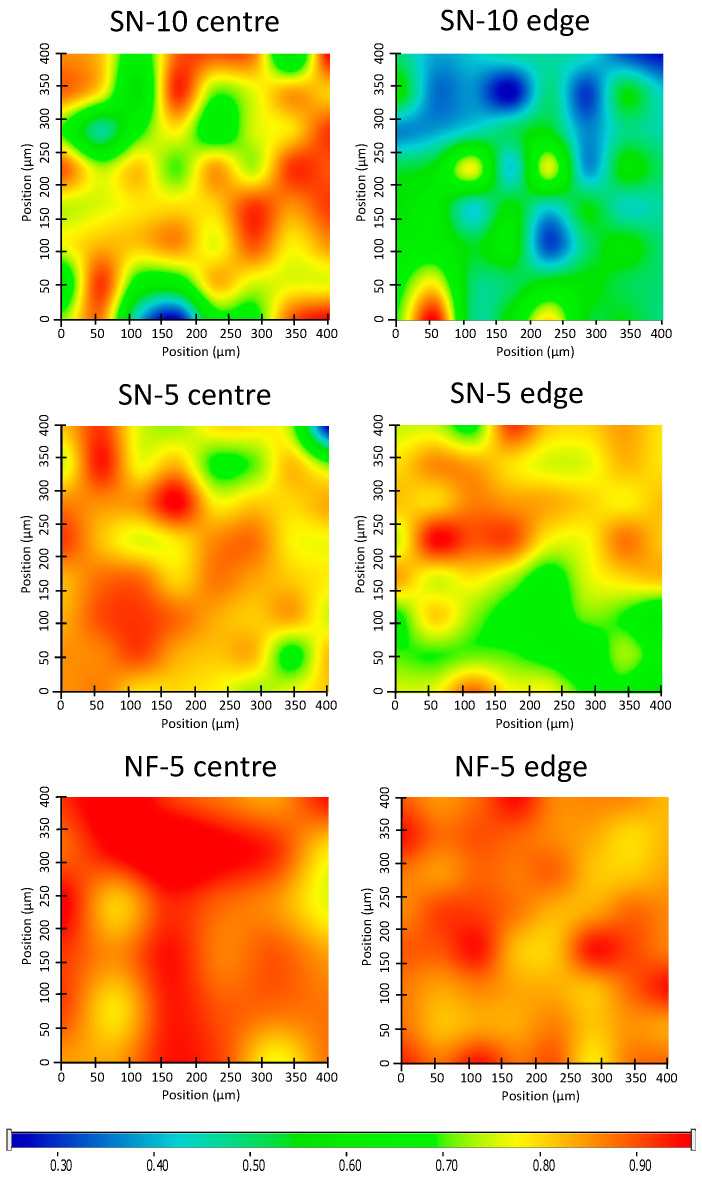
Raman chemical maps of nanofiber mats indicating the distribution of ciprofloxacin in the samples. Raman maps were captured on specimens collected from the center and edge of the collector. The drug distributions of the nanofiber mats were more homogeneous for nozzle-free electrospinning than for the method using a nozzle.

**Figure 6 pharmaceutics-14-01559-f006:**
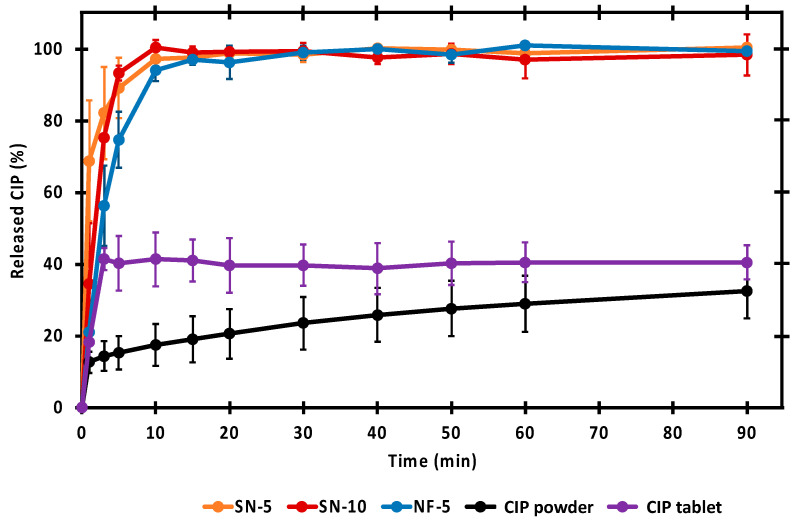
In vitro dissolution of ciprofloxacin (CIP) from the nanofibrous samples, CIP powder, and tablet. Nearly the total amount of CIP was rapidly released from the nanofibers prepared via single-nozzle (SN-5 and SN-10) and nozzle-free (NF-5) electrospinning.

**Figure 7 pharmaceutics-14-01559-f007:**
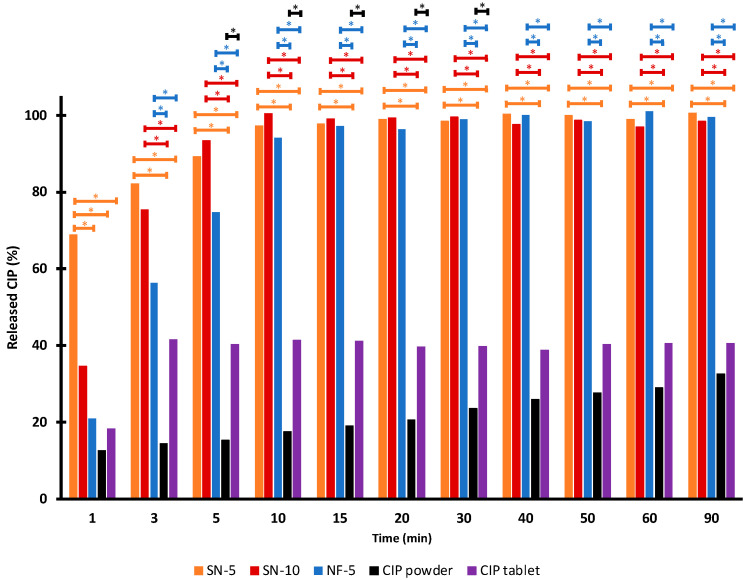
Average ciprofloxacin (CIP) release at the sampling points. * means significantly different (*p* < 0.01) values calculated using one-way ANOVA, with post hoc Tukey HSD testing of each sample, pair by pair.

**Figure 8 pharmaceutics-14-01559-f008:**
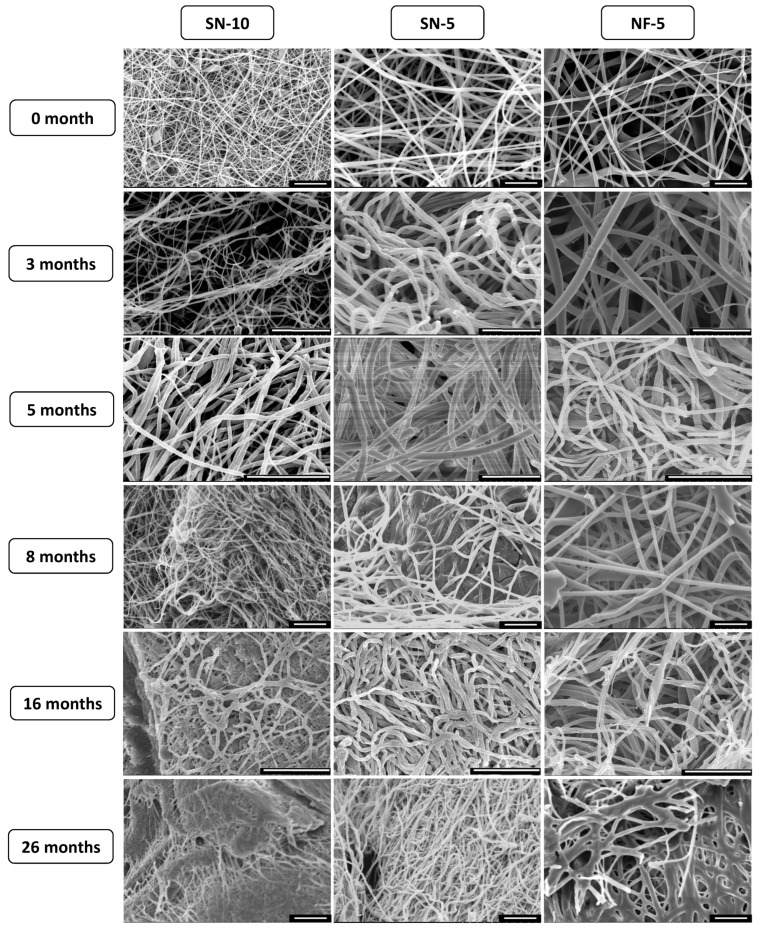
SEM images of the ciprofloxacin-loaded nanofibers taken during the 26-month stability study. The morphology of the nozzle-free nanofiber sample (NF-5) remained stable over the 26 months. For the nanofibers produced using a nozzle (SN-10 and SN-5), aging of the structure was observed at 16 and 26 months. The scale bar is 20 µm on every microscopic image.

**Figure 9 pharmaceutics-14-01559-f009:**
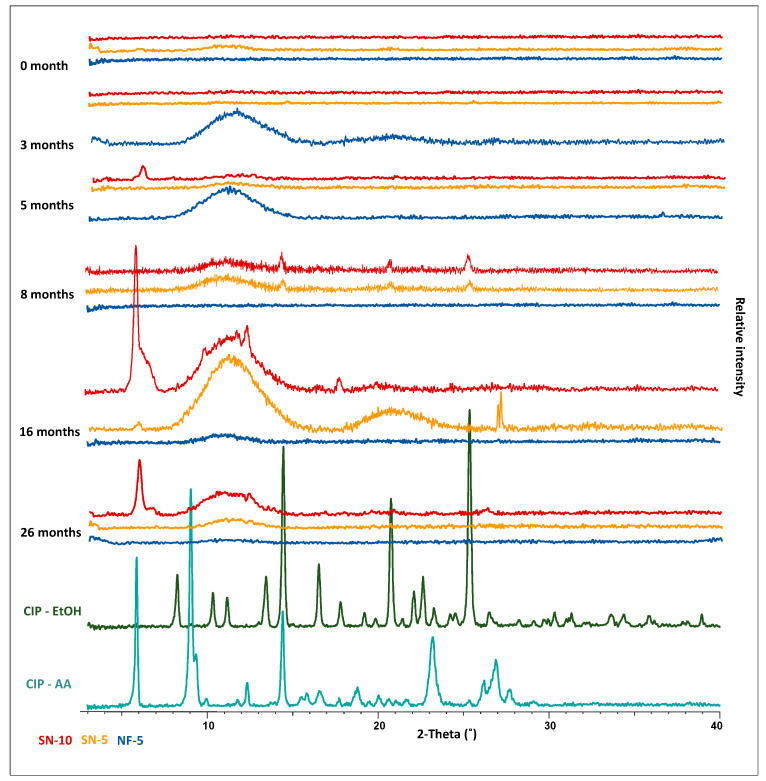
XRPD results of the 26-month stability study. The nanofiber formulations measured at different time points are colored as follows: red and yellow for single-nozzle electrospinning (SN-10 and SN-5), and blue for nozzle-free electrospinning (NF-5). XRPD patterns detected from crystallized ciprofloxacin from ethanol (CIP-EtOH) and acetic acid (CIP-AA) are green. In terms of crystallinity, better stability was observed for samples produced using nozzle-free electrospinning, as the drug was amorphous throughout the 26 months. In the case of nozzle-based formulations, the ciprofloxacin re-crystallized by the 5th and 8th months.

**Figure 10 pharmaceutics-14-01559-f010:**
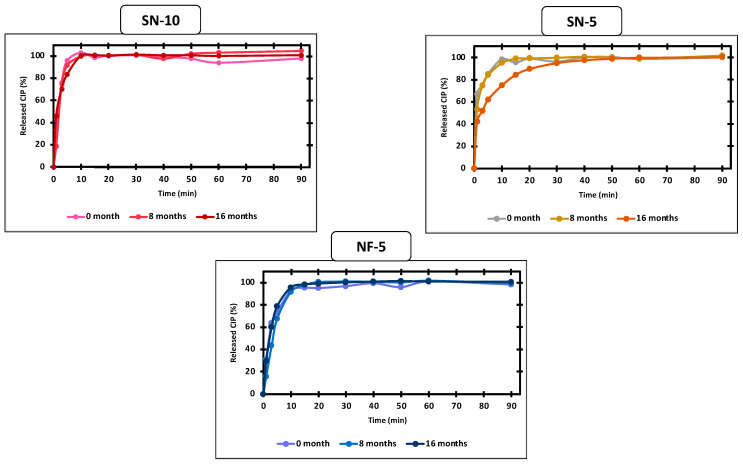
Comparison of in vitro ciprofloxacin (CIP) release of nanofibers produced using single-nozzle (SN-10 and SN-5) and nozzle-free (NF-5) electrospinning after 0, 8, and 16 months of storage. After 8 months of storage, there was no change in the drug release of any samples, but at the age of 16 months, the SN-5 sample showed an approximately three-fold slower release rate.

**Figure 11 pharmaceutics-14-01559-f011:**
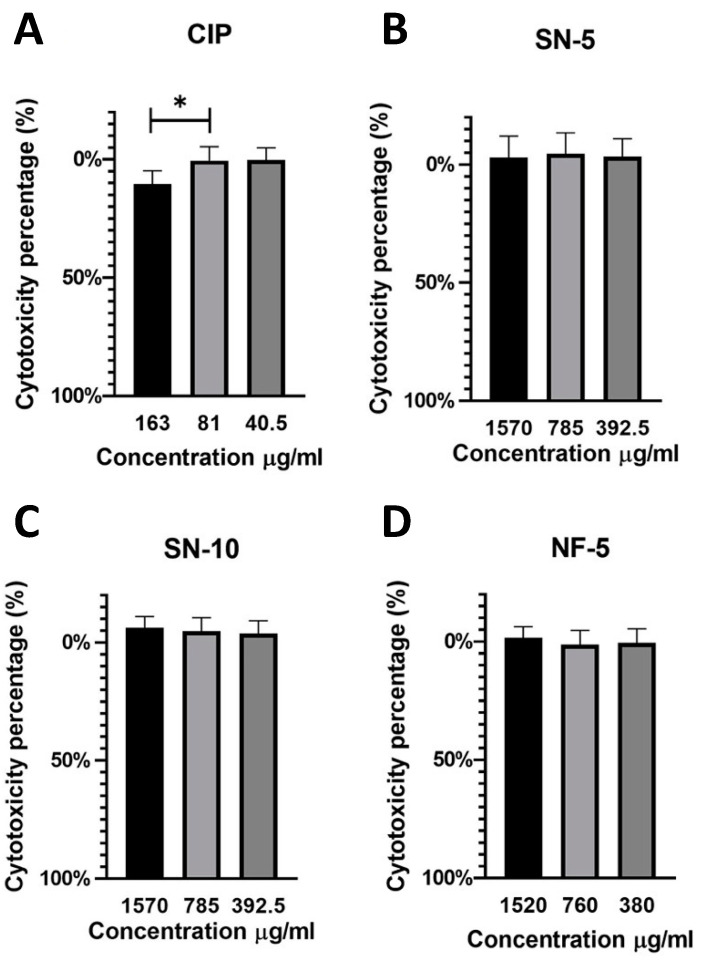
In vitro cytotoxicity measurement of the following solutions. (**A**) untreated ciprofloxacin (CIP) powder, (**B**) 5 w% CIP nanofibers produced using single-nozzle electrospinning, (**C**) 10 w% CIP nanofibers prepared using single-nozzle electrospinning, and (**D**) 5 w% CIP nanofibers produced using nozzle-free electrospinning. The solution of the untreated CIP powder was found to be cytotoxic at 163 µg/mL, but the solutions of different nanofibers were not cytotoxic in any of the measured concentrations (* *p* < 0.05).

**Table 1 pharmaceutics-14-01559-t001:** Stability studies of electrospun nanofibers.

Time Interval of the Study	Tested Properties of the Nanofibers				
	Morphology	Crystallinity	Drug Loading	In vitro Drug Release	Reference
1 month	×	×			[49]
1 month	×		×		[50]
45 days			×		[51]
3 months	×				[52]
3 months	×		×		[53]
3 months		×		×	[54,55]
4 months		×			[46]
6 months		×			[42,56]
8 months		×			[44,57]
10 months		×			[45]
12 months		×			[58]
12 months			×		[59]
12 months		×		×	[60]

**Table 2 pharmaceutics-14-01559-t002:** The composition and preparation procedures of each sample. Abbreviations: CIP—ciprofloxacin, PVP—polyvinylpyrrolidone, ES—electrospinning.

Sample			Volume Ratio				
	CIP Solution in Acetic Acid (mg/mL)	PVP Solution in Ethanol (*w*/*v*%)	PVP	CIP	Viscosity of the ES Solution (mPa s)	ES Setup	CIP-Loading (wt%)
SN-5	20	10	4	1	79 ± 2	single-nozzle	5
SN-10	20	5	4	1	336 ± 19	single-nozzle	10
NF-5	20	10	4	1	79 ± 2	nozzle-free	5
NF-10	20	5	4	1	336 ± 19	nozzle-free	10

**Table 3 pharmaceutics-14-01559-t003:** Fiber morphologies and average diameters of different formulations.

Electrospinning Setup	Sample	Fiber Morphology		Av Diameter ± S.D. (nm)
single-nozzle	SN-5	continuous, smooth-surfaced	mainly uniform, some beads	323 ± 51
	SN-10	continuous, smooth-surfaced	large number of beads and bags	735 ± 91
nozzle-free	NF-5	continuous, smooth-surfaced	thicker and thinner fibers	1167 ± 415
	NF-10	merged, rough-surfaced	more film than nanofiber	unmeasurable

**Table 4 pharmaceutics-14-01559-t004:** Drug-loading (DL) and entrapment efficiency (EE) of different formulations from the centers and the edges of the nanofiber mats. Abbreviations: ES—electrospinning, CIP—ciprofloxacin, S.D.—standard deviation.

ES Setup	Sample	Theoretical CIP Content (%)	DL (%) ± S.D.	EE (%) ± S.D.
single-nozzle	SN-10 center	10	8.72 ± 1.42	87.2 ± 14.2
	SN-10 edge	10	7.01 ± 1.64	70.1 ± 16.4
	SN-5 center	5	3.70 ± 1.62	73.9 ± 32.5
	SN-5 edge	5	3.36 ± 0.61	67.1 ± 12.1
nozzle-free	NF-5 center	5	4.43 ± 0.47	88.7 ± 14.2
	NF-5 edge	5	4.67 ± 0.75	93.3 ± 15.0

**Table 5 pharmaceutics-14-01559-t005:** Regression coefficient (R^2^) values of the drug-release models and the measured ciprofloxacin (CIP) release from the samples.

Model	SN-5	SN-10	NF-5	CIP Powder	CIP Tablet
Zero order	0.5297	0.7317	0.7874	0.7510	0.5636
First order	0.9352	0.8809	0.9870	0.8062	0.5794
Higuchi	0.8103	0.9327	0.9491	0.9148	0.8243
Hixson–Crowell	0.8061	0.9876	0.9437	0.7880	0.5745
Korsmeyer–Peppas	0.9994	0.8657	0.9836	0.9033	0.7334

**Table 6 pharmaceutics-14-01559-t006:** Average fiber diameters of nanofibers produced using nozzle-based (SN) and nozzle-free (NF) electrospinning during the 26-month stability test.

	Average Fiber Diameter ± Standard Deviation (nm)		
Time	SN-10	SN-5	NF-5
0 month	323 ± 51	735 ± 91	1167 ± 415
3 months	328 ± 84	739 ± 105	1195 ± 464
5 months	345 ± 108	866 ± 233	1159 ± 406
8 months	304 ± 53	805 ± 133	1418 ± 343
16 months	434 ± 69	783 ± 162	1444 ± 478
26 months	442 ± 89	707 ± 185	2148 ± 853

## Data Availability

The datasets used and/or analyzed are available from the corresponding author on reasonable request.

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
