# Peer review of "Comparison of Nozzle-Based and Nozzle-Free Electrospinning for Preparation of Fast-Dissolving Nanofibers Loaded with Ciprofloxacin"

_pharmaceutics, 2022, doi:10.3390/pharmaceutics14081559_

Round 1
Reviewer 1 Report
The authors scaled up the electrospinning method with a nozzle free approach, also compared the properties of nanofibers of the poorly water-soluble ciprofloxacin and polyvinylpyrrolidone produced with these two methods.The results showed that nozzle free approach not only offers high throughput, but also improves homogeneity and enhanced stability of nanofiber mats.There are several issues should be addressed.
1. From the Figure 2 and 3, it can be seen clearly that the nanofibers using conventional espining is more uniform than the nozzle free spinning. How could the mats with non-uniformity in size offer better homogeneity? The authors might give more explanations.
2. Different size range of nanofibers might be obtained using nozzle free approach, it might be beneficial to cell growth and improve the stability, therefore, the authors might conduct cell growth test and also test the effect of ciprofloxacin.
Author Response
Peer Reviewer 1
Comments and Suggestions for Authors
The authors scaled up the electrospinning method with a nozzle free approach, also compared the properties of nanofibers of the poorly water-soluble ciprofloxacin and polyvinylpyrrolidone produced with these two methods. The results showed that nozzle free approach not only offers high throughput, but also improves homogeneity and enhanced stability of nanofiber mats. There are several issues should be addressed.
A: Thank you very much for your comments and questions. Below are listed the responses to your queries.
- From the Figure 2 and 3, it can be seen clearly that the nanofibers using conventional espining is more uniform than the nozzle free spinning. How could the mats with non-uniformity in size offer better homogeneity? The authors might give more explanations.
A: Thank you for the question. The diameter of the laser beam used for the Raman measurement is thicker than the individual nanofibers, so the difference in nanofiber diameter (i.e. the non-uniformity) does not show up in the Raman map. With nozzle-free electrospinning, a more uniform distribution of the fibers on the collector was ensured, resulting in better homogeneity.
- Different size range of nanofibers might be obtained using nozzle free approach, it might be beneficial to cell growth and improve the stability, therefore, the authors might conduct cell growth test and also test the effect of ciprofloxacin.
A: We would like to express our gratitude to the Reviewer for bringing this to our attention, as this technique would considerably improve the manuscript's quality. This would be a particularly interesting proposal for nanofibers intended for topical use but our work (in line with the scope of the special issue) is focused on oral administration. So, the use of the solid nanofiber mat without dissolution would be irrelevant in this study. Therefore, a cell proliferation test was performed as described below. However, the test has not been included in the is article due to the difficulty of interpreting the highly varying data.
Methods:
CaCo-2 cells were seeded in the concentration of 4 x 10^3/well in 96 wells plate. The next day the culture medium was replenished, with a new medium and was supplemented with the sample’s concentration used in cytotoxicity test. The final volume of the medium containing the samples was 200 ml. Cells were incubated for 4 days at 37 °C, with a 5% CO2 concentration. After the incubation period, the measurement was conducted as described in ‘In vitro cytotoxicity’ section.
Results and Discussion:
The study started with a relatively small number of cells and then observed the proliferation of cells treated with different samples and compared to the cell control. Cell proliferation was indeed improved by both PVP and SN-5 at different concentrations, where higher cell proliferation was observed than in the tissue control. However, no correlation was found between the concentration and the magnitude of cell growth. Interestingly, in the cases of NF-5 and SN-10, cell growth was well below the control and similar to the pure CIP solution. Such a difference between NF-5 and SN-5 is unlikely to be due to the electrospinning method, as they were applied to the cells in a dissolved state. The results are potentially variable due to the nature of the cellular assays, despite that the experiment was performed in 4 parallel
measurements.
Figure 1. In vitro cell proliferation test
In the future, we intend to formulate nanofibers targeting other routes of administration, in which the proposed measurements will be certainly included in the investigations of nanofibers for topical use.
We would like to thank you again. We really appreciate your revision, which helped us to substantially improve the manuscript.

Author Response
Peer Reviewer 2
I have been asked to review a paper entitled ‘Comparison of nozzle-based and nozzle-free electrospinning for preparation of fast-dissolving nanofibers loaded with ciprofloxacin’. Herein some selective comments towards the authors that might improve this work.
A: Thank you very much for your comments and questions. Below are listed all responses and modifications made in the paper, according to your suggestions. You can find the added parts with orange color in the manuscript.
Abstract
It is important to improve the English language in the Abstract and the manuscript as whole.
A: Thank you for the suggestion. The English language issues have been corrected.
The opening statements of this manuscript (lines 15 to 20) should be improved. Try to use topic-related (i.e. study-related) statements instead.
Thank you for your comment. The abstract has been revised and rewritten. See below.
Abstract: The study aimed to prepare ciprofloxacin-loaded polyvinylpyrrolidone electrospun nanofibers for oral drug delivery by a conventional nozzle-based and a lab-built nozzle-free electrospinning equipment. To produce nanofibers, electrospinning is the most often used process. However, from the industry's point of view, conventional electrospinning does not have sufficiently high productivity. By omitting the nozzle, productivity can be increased, so developing nozzle-free processes is worthwhile. In this study, solutions of ciprofloxacin and polyvinylpyrrolidone were electrospun under similar conditions by both single-nozzle and nozzle-free methods. The two electrospinning methods were compared by investigating the morphological and physicochemical properties, homogeneity, in vitro drug release, and cytotoxicity. The stability of the nanofibers was monitored from different aspects in a 26-month stability study. The results showed that the use of the nozzle-free electrospinning was advantageous due to a higher throughput, improved homogeneity, and enhanced stability of nanofiber mats, compared to the nozzle-based method. Nevertheless, fast dissolving nanofibers loaded with poorly water-soluble ciprofloxacin were produced by both electrospinning method. The beneficial properties of these nanofibers can be exploited in innovative drug development, e.g. nanofibers can be formulated into orodispersible films or per os tablets.
Introduction
It is preferable not to use words like ‘we or our’.
A: Thank you for the suggestion. The sentences in 74-76. and 82-83. lines were changed.
[74-76] The latter concept is also the basis of the nozzle-free ES device, in which nanofibers are formed from the surface of a rotating cylinder.
[82-83] Similarly, in a previous study done by the research group, the solubility of CIP in pH 7.4 PBS was found to be 0.099 mg/mL.
The PVP section should be more descriptive.
A: Thank you for the suggestion. We have revised the section.
Polyvinylpyrrolidone (PVP) or povidone is one of the most utilized pharmaceutical polymers. It can be found as a binder in traditional pharmaceutical products such as tablets, but it is also widely used in nanotechnology as a drug delivery polymer. PVP is in favor because it is a non-toxic, biocompatible, biodegradable polymer that is easy to handle [41]. Besides other nanosystems, PVP is a commonly used polymer for nanofibers developed as drug carriers [42–48]. PVP is a water-soluble polymer, which makes it well suited for use in orodispersible or rapid release formulations.
Table 1 should go down in the results and discussion section.
A: Thank you for your comment. The table is a literature review, a summary of stability tests done by other research groups. Because of the strong literature aspect, we think it belongs in the introduction.
The use of 26-week instead of 24-week (6 months) is odd.
A: This might be a misunderstanding as ‘26 weeks’ is not used in the manuscript. The stability study was performed over 26 months.
Material and methods
2.3. Use Figure 1.A instead of Figure 1.
A: Thank you. It has been corrected.
One of the major limitation in this study is the use of flat collector in one system and a rotating collector in the other. The former will produce random oriented fibers while the latter will produce aligned fibers.
A: Thank you for the comment. During the nozzle-free spinning, the collector was rotating at a low speed, so the difference mentioned was not observed. Both Figure 2 and Figure 3 SEM images show that randomly oriented nanofibers could be produced by both ES methods.
2.4. Use Figure 1.B instead of Figure 1.
A: Thank you. It has been corrected.
2.6.5. Please check the accuracy of equations 1 and 2
How did the authors calculated
???????? ???? ?? ???
???????? ???? ???????
?ℎ????????? ???? ???????
A: Thank you for the comment. To clarify the equations, we revised them and added some explanations, as follows.
DL and EE of the nanofibrous formulations were calculated according to the following equations:
(1)
The nanofiber weight was obtained by weighing the 0.5x0.5 cm specimens. After the dissolution of the specimens in pH 7.4 PBS, the measured mass of CIP was calculated from the UV absorption of the drug.
(2)
Measured CIP concentration was calculated from the UV absorption of CIP after weighing and dissolving the specimens in pH 7.4 PBS. In addition, the initial CIP concentration in the electrospinning solution was considered as total CIP content.
Why the authors choose to measure the EE and DL by UV and the release by HPLC? Why not both using HPLC?
A: Thank you for the question. UV spectroscopy is a simple, fast, and cheap method to determine the concentration of an API and therefore it is often used. The absorbance of PVP does not overlap with the ciprofloxacin’s, so UV spectroscopy can be used with confidence in our case [1–5].
A commercially available tablet was also used as a reference in the release studies. Due to the large number of excipients, separation of the components was essential in this case, therefore we used HPLC.
- Ajmal, G.; Bonde, G.V.; Thokala, S.; Mittal, P.; Khan, G.; Singh, J.; Pandey, V.K.; Mishra, B. Ciprofloxacin HCl and Quercetin Functionalized Electrospun Nanofiber Membrane: Fabrication and Its Evaluation in Full Thickness Wound Healing. Artificial Cells, Nanomedicine, and Biotechnology 2019, 47, 228–240, doi:10.1080/21691401.2018.1548475.
- Aytac, Z.; Ipek, S.; Erol, I.; Durgun, E.; Uyar, T. Fast-Dissolving Electrospun Gelatin Nanofibers Encapsulating Ciprofloxacin/Cyclodextrin Inclusion Complex. Colloids and Surfaces B: Biointerfaces 2019, 178, 129–136, doi:10.1016/j.colsurfb.2019.02.059.
- Ayati Najafabadi, S.A.; Shirazaki, P.; Zargar Kharazi, A.; Varshosaz, J.; Tahriri, M.; Tayebi, L. Evaluation of Sustained Ciprofloxacin Release of Biodegradable Electrospun Gelatin/Poly(Glycerol Sebacate) Mat Membranes for Wound Dressing Applications. Asia-Pac J Chem Eng 2018, 13, e2255, doi:10.1002/apj.2255.
- KyzioÅ‚, A.; Michna, J.; Moreno, I.; Gamez, E.; Irusta, S. Preparation and Characterization of Electrospun Alginate Nanofibers Loaded with Ciprofloxacin Hydrochloride. European Polymer Journal 2017, 96, 350–360, doi:10.1016/j.eurpolymj.2017.09.020.
- Modgill, V.; Garg, T.; Goyal, A.K.; Rath, G. Permeability Study of Ciprofloxacin from Ultra-Thin Nanofibrous Film through Various Mucosal Membranes. Artificial Cells, Nanomedicine, and Biotechnology 2016, 44, 122–127, doi:10.3109/21691401.2014.924007.
2.6.6. Is the HPLC method developed or adapted from a previous study?
A: Thank you for the question. The HPLC method used to quantify ciprofloxacin was developed.
Please show the HPLC peak (drug separation) and the standard curve; and add them in the Supplementary Section.
A: Thank you for your comment, a Supplementary Section has been created. The linearity of the calibration curve was tested in 1-1000 μg/mL concentration range, by 8 concentration points. The plotted calibration curve and an example chromatogram of CIP was provided in the supplementary material (Figure S2 and S3). The LOD and LOQ of CIP was 135 ng/mL and 411 ng/mL, respectively.
In the Materials section, the authors mentioned and acidic PBS (2.8), where did they use it (under which section)?
A: Apologies for the typo, pH 2.8 PBS was applied as eluent for the HPLC method. We have corrected this mistake in the manuscript.
2.6.9. Why did the authors choose Caco-2 cell lines? How did they correlate this cell lines to the cytotoxicity (safety) study? Again, it is an odd choice.
A: Thank you for your questions. The study focused on nanofiber formulations for per os administration. Medicines that have entered the digestive tract spend most of their time in the intestines - if they are not absorbed sooner. Ciprofloxacin is absorbed mostly from the small intestine. Caco-2 cell line can model the small intestine as well as the colon [6,7]. For these reasons, we investigated the cytotoxicity of the nanofiber solutions on Caco-2 cells.
- Fröhlich, E. Comparison of Conventional and Advanced in Vitro Models in the Toxicity Testing of Nanoparticles. Artificial Cells, Nanomedicine, and Biotechnology 2018, 46, 1091–1107, doi:10.1080/21691401.2018.1479709.
- Sambruy, Y., Ferruzza, S., Ranaldi, G., De Angelis, I. Intestinal Cell Culture Models: Applications in Toxicology and Pharmacology. Cell Biology and Toxicology, 2001; 17, 301–317., doi:10.1023/a:1012533316609
The MTT assay should be written properly. How the author added the MTT solution without aspirating the consumed medium?
A: Thank you for the question. According to the cytotoxicity measurement methods, it is possible to add the MTT solution to the existing media. In this scenario, it's essential to ensure that each sample has the same amount of existing media.
Please find reference for the method:
https://www.abcam.com/kits/mtt-assay-protocol
https://www.sigmaaldrich.com/HU/en/technical-documents/protocol/cell-culture-and-cell-culture-analysis/cell-counting-and-health-analysis/cell-proliferation-kit-i-mtt
Why did the use sodium dodecyl sulfate solution? If it was to dissolve the formed formazan crystal, it should be better replaced with DMSO or isopropanol.
A: Thank you for the comment. In our lab, sodium dodecyl-sulfate has been utilized for approximately ten years. We were able to publish a number of publications in peer-reviewed journals using this strategy. The sodium dodecyl-sulfate besides DMSO or isopropanol is capable of dissolving the formazan crystal that has formed.
Please find reference for the method:
- Triterpenes and Phenolic Compounds from Euphorbia deightonii with Antiviral Activity against Herpes Simplex Virus Type-2, Muhammad et al., Plants 2022
- Physico-Chemical, In Vitro and Ex Vivo Characterization of Meloxicam Potassium-Cyclodextrin Nanospheres, Varga et al., Pharmaceutics 2021
Results
This section should be names Results and Discussion
A: Thank you for drawing attention to this. It has been corrected.
It is unusual that the 10% PVP showed bead. Something is wrong here. By checking the study parameters, it seems that the parameters are right that’s why it is weird not to have proper fibers at the concentration especially by using the PVP MW. Please clarify this and compare the interpretation to previous studies using similar PVP MW.
A: Thank you for the comment. In our previous experiments we used the same PVP MW but with lower CIP content. In that case, proper fibers could be produced. To achieve higher concentration, we had to change the solvent of CIP from chloroform to acetic acid. It is likely that increasing the CIP loading or changing the solvent had a negative effect on the morphology of the fibers.
The previous paper published with CIP and PVP MW can be find here:
Uhljar, L.É.; Kan, S.Y.; Radacsi, N.; Koutsos, V.; Szabó-Révész, P.; Ambrus, R. In Vitro Drug Release, Permeability, and Structural Test of Ciprofloxacin-Loaded Nanofibers. Pharmaceutics 2021, 13, 556, doi:10.3390/pharmaceutics13040556.
If the 10% PVP was too viscus, why didn’t the authors choose 7 or 8% instead?
A: Thank you for your question. We believe that it could be useful to publish a composition where fiber formation was successful with one production method and not with the other method. This result may be helpful to other researchers, so we have not attempted to reduce the concentration. Fibers containing 5% CIP can be compared between the two methods, while the effect of concentration can be investigated on the samples made by the single nozzle electrospinning.
Figure 2 and 3 should be labelled, there are several images. Use numbers or letters in the top corner of each image, then describe the letter in the figure legend. Scale bar doesn’t show the measurement.
A: Thank you for the suggestion. You can see the new figures here.
Figure 2. SEM images and diameter distributions of nanofibrous samples produced by nozzle-based electrospinning. The scale bar is 20 µm on every microscopic image. A1. The average fiber diameter was 323 ± 51 nm in the case of nanofibers containing 5 w% ciprofloxacin (SN-5). The size distribution of nanofibers was monodisperse. A2 and A3. The morphology of the fibers was mainly uniform, but some beads formed. B1. In the case of the nanofibers containing 10 w% ciprofloxacin (SN-10), the average fiber diameter was significantly larger (735 ± 91 nm). The size distribution had bell-shape. B2 and B3. Beside the numerous beads, large sack-shaped formations also appeared proving the imperfect electrospinning process.
Figure 3. SEM images and diameter distributions of nanofibrous samples produced by nozzle-free electrospinning. The scale bar is 10 µm on all microscopic images. A1. In the case of the nanofibers containing 5 w% ciprofloxacin (NF-5), the free-surface electrospinning was successful. The average fiber diameter was 1167 ± 415 nm, but the size distribution was polydisperse and strongly skewed left. A2 and A3. The fibers were mainly individual and had smooth surface. B2 and B3. The nozzle-free electrospinning of 10 w% ciprofloxacin containing nanofibers (NF‑10) was impractical while instead of separated fibers merged fiber balls and polymer film were prepared. B1. Thus, the fiber size was not measurable, and the distribution was not interpretable.
DSC and XRD results are not well stated.
A: Thank you for your comment. The section is rewritten now as following.
It is known that during the ES process the crystal structure of the active substance may change, in most cases amorphizing [3,60]. Therefore, nanofibers can be considered as amorphous solid dispersions. Two types of experiments, DSC and XRPD, were performed to prove that both ES methods could produce amorphous solid dispersions. Figure 4A presents the DSC thermograms of CIP powder, physical mixtures (PM‑5 and PM‑10), and samples made by nozzle-based ES (SN‑5 and SN‑10) and nozzle-free ES (NF‑5). The sharp endothermic peak at 275 °C for CIP powder and physical mixtures represents the melting point of crystalline CIP. As the size of the peak depending on the amount of the crystalline material, the peak of the 5% CIP-loaded sample (PM‑5) was about half the intensity of the peak of the 10% CIP-containing sample (PM‑10). However, the peak did not appear in the thermograms of nanofibers because of the absence of a measurable melting point. Thus, all nanofibrous samples contained amorphized CIP regardless of the preparation method, and all the produced nanofibers can be considered as amorphous solid dispersions.
Beside DSC measurements, the change in the crystallinity of CIP was studied by XRPD study (Figure 4B). The CIP powder showed high crystallinity as long, sharp peaks appeared on its diffractogram at around 2-Theta = 5.8, 9.1, 14.5, and 23.2°. In the case of PM-10, there were characteristic CIP peaks and a wide peak typical of PVP between 7.5 and 15.2°. In contrast, the diffractograms of the nanofibrous samples (SN-5, SN-10, and NF-5) indicated amorphous API. Hence, the sample PM-10 contained crystalline API as a proof that the amorphization had occurred during the production and not because of any polymer-drug interactions.
The homogeneity problem might be because of the 24-hour stirring which might allowed the ethanol to evaporate and increased the viscosity of the polymer solution. Mixing the polymer solution with the acidic drug solution changed the pH and might affect the spinning. Why the solution pH was not measure?
A: Thank you! The vials we used for mixing were well closed to minimize evaporation. For solution preparation, the PVP and CIP solutions were mixed in a 4:1 ratio in all cases, so there should not be much difference in the pH of the electrospinning solutions, and therefore it was not considered necessary to determine it.
The stat differences for the release study (Table 5) should not be in Table format. Authors should be use a different bar chart for example to present this well.
A: Thank you for the suggestion. We have replaced Table 5 with Figure 7.
Figure 7. Average ciprofloxacin (CIP) release at the sampling points. * means significantly different (p < 0.01) values calculated by one-way ANOVA with post-hoc Tukey HSD testing each sample pair by pair.
The stability study is not a realistic one as the fibers were stored in the air tight conditions rather than the usual storage conditions (Temp & R/H). Please address!
A: Thank you for the comment. A sentence was added at the end of the 3.7 section.
In the case of NF-5, the CIP remained amorphous after 26 months, and consequently there was no change in in vitro dissolution. Overall, in terms of drug release, the nanofibers prepared with nozzle-free ES was found to be more stable but the limiting factor that the samples were stored in a desiccator should be taken into consideration.
Why the conc of Cipro in the cytotoxicity study is different from the formulations?
A: Thank you for your question. Ciprofloxacin is poorly soluble in water, which can be improved by the solubilizing agents of the media. However, 163 mg/mL was the maximum CIP concentration we could achieve. It is important to note that Figure 9 shows the concentrations of the drug-loaded nanofibrous samples. The highest CIP concentrations are the following: 78,5 µg/mL in the case of SN-5, 157 µg/mL in the case of SN-10, and 76 µg/mL in the case of NF 5. Thus, it can be seen that we have worked with comparable amounts and relatively narrow concentration ranges considering the CIP.
The author should had considered microbiology assay (zone of inhibition) better than the cytotoxicity. Does the antibiotic retained its efficiency? This is another major limitation.
A: Thank you for the suggestion and the question. Our aim was to produce a nanofiber developed for per os administration. The release of the ciprofloxacin from the nanofibers is rapid, therefore it is assumed to retain its antibiotic activity.
Discussion
There is not discussion!
It is highly recommended to combine both results and discussion as authors didn’t discuss their results.
A: Thank you for drawing attention to this, again. Paragraph 3 contains the discussion alongside the results, so we have rewritten the subtitle to ‘3. Results and Discussion’.
Conclusion
Needs to be improved and describe the results better.
A: Thank you for the comment. The conclusion is revised and rewritten now.
Nanofibers as amorphous solid dispersions can be formulated as orodisperse films or oral medicines while with the suitable polymer, fast disintegration and rapid release can be achieved. This paper presented the production and investigation of CIP-loaded PVP nanofibers prepared by nozzle-based and nozzle-free ES methods. Nanofibers with 5% and 10% CIP concentration were fabricated by the conventional single-nozzle ES. By the nozzle-free method, 5% CIP-loaded nanofibrous samples were produced from the same electrospinning solutions. Comparing the nanofibers, we found that the preparation method had no influence on the drug carriers as amorphous solid dispersions because both methods could amorphize the CIP. Also, the presence or absence of the nozzle had no effect on the in vitro drug release, as the dissolution of the CIP was complete and fast from the nanofibers and occurred according to Korsmeyer-Peppas model or first-order kinetics. Finally, none of the prepared nanofibrous samples were cytotoxic according to the MTT test on CaCo-2 cell lines. This study may suggest further processing steps, which might include the production of an orodisperse film by pressing or production of per os drugs by filling the fibers into gelatin capsules.
However, it can be concluded that nozzle-free ES has several advantages. These are, based on the studies performed, a more homogeneous distribution of the active ingredient within the nanofiber mat, a higher encapsulation efficiency, and a longer stability. An additional advantage may be the increased productivity due to the fact that ES from the free solution surface allows the formation of several Taylor cones at the same time. On the other hand, this also leads to the disadvantage of the nozzle-free method, which is the wide variability in fiber diameters.
The study also focused on the stability of nanofibers. In this context, SEM, XRPD and in vitro release studies were carried out, which led to two types of conclusions. The first conclusion was that aged morphology, fusion of the fibers or partial recrystallisation of the CIP did not affect the in vitro drug release. Based on these results, it is important to check for drug release in all stability studies of nanofibers. As a second conclusion, the stability results suggested that NF-5 was sufficiently stable for 16 months in terms of morphology and 26 months in terms of amorphousness and in vitro release. In contrast, the morphology of the SN-5 sample remained unchanged up to 26 months, but the CIP started to recrystallize at 8 months and its release slowed down significantly at 16 months. Thus, the nanofibers prepared with nozzle-free ES was found to be more stable during the 26-month storage in a desiccator.
Overall, nanofiber production by nozzle-free ES equipment can not only provide a good alternative to the nozzle-based ES but also produces nanofibers with improved properties. Thus, it could be worthwhile to develop the method further and use the resulting nanofibers as drug delivery systems.
Again, thank you for your time and all the questions and comments. Hopefully we gave a satisfactory answer to each of your questions.

Reviewer 3 Report
"Comparison of nozzle-based and nozzle-free electrospinning for preparation of fast-dissolving nanofibers loaded with ciprofloxacin" is a well-written and quite interesting manuscript dealing with the important development of drug-loaded biodegradable nanofibers. Indeed, this work will attract the attention of the Pharmaceutics audience thanks to important findings and the very good design of the research. However, some minor changes are possibly needed:
1) The formation of beads for SN-10 and NF-5 (and I am sure for NF-10, although there were even no fibers at all). Were you considering adjusting the voltage, solution rate, and other parameters to minimize the effect?
2) Raman measurements. The authors provided the mapping of CIP, but obviously that this is the result of some distribution of specific peaks intergrations. However, the authors have not shown the origin of those peaks. Please include in the paper of Supp Info the description of the methodology and Raman spectra
3) Please consider using X-ray Photoelectron Spectroscopy and the special fitting methodology, to define CIP and PVP percentage, e.g. as described in https://www.mdpi.com/1999-4923/14/4/724
Author Response
Peer Reviewer 3
Comments and Suggestions for Authors
"Comparison of nozzle-based and nozzle-free electrospinning for preparation of fast-dissolving nanofibers loaded with ciprofloxacin" is a well-written and quite interesting manuscript dealing with the important development of drug-loaded biodegradable nanofibers. Indeed, this work will attract the attention of the Pharmaceutics audience thanks to important findings and the very good design of the research.
A: Thank you very much for your kind words. We are glad that You are satisfied with our work. Below are listed the responses of your comments and questions. You can find the added parts with violet color in the manuscript.
However, some minor changes are possibly needed:
1) The formation of beads for SN-10 and NF-5 (and I am sure for NF-10, although there were even no fibers at all). Were you considering adjusting the voltage, solution rate, and other parameters to minimize the effect?
A: Thank you for your question. The configuration parameters described in the article are the result of an optimization process. We have varied the voltage, the nozzle-to-collector distance, and the flow rate to find the optimal parameters. However, with these parameters the best morphology could be achieved, unfortunately we could not produce more uniform and beadless fibers.
2) Raman measurements. The authors provided the mapping of CIP, but obviously that this is the result of some distribution of specific peaks integrations. However, the authors have not shown the origin of those peaks. Please include in the paper of Supp Info the description of the methodology and Raman spectra
A: Thank you for your comment, the origin of peaks and the methodology of constructing Raman maps was added to the supplementary section (Figure S1).
Raman spectral profiling
Figure S1. Raman spectra of investigated fiber mat and initial components, indicating the unique spectral range of CIP applied for constructing Raman chemical maps.
For investigating the distribution of CIP in the fiber mats the spectral range from 1650 to 1550 cm-1 was selected for profiling to determine the frequency of occurrence of CIP in the electrospun specimens. The selected spectral region contained two intense signals centered at 1615 cm−1 because of the aromatic ring stretching mode ν(C=C) of the quinolone ring system coupled with the carbonyl stretching mode ν(C=O) at 1589 cm-1 were suitable for analysis, as in this spectral region no overlapping band of PVP could be found.
3) Please consider using X-ray Photoelectron Spectroscopy and the special fitting methodology, to define CIP and PVP percentage, e.g. as described in https://www.mdpi.com/1999-4923/14/4/724
A: Thank you very much for your suggestion. Unfortunately, we do not have the appropriate instrumentation now, but we will keep your suggestion in mind and in the future, we will try to carry out such measurements in cooperation.
We would like to thank you again. We really appreciate your comments and suggestions.

Round 2
Reviewer 1 Report
The revision is satisfying.
Author Response
Dear Colleague,
Thank you very much once again for your time and ideas for improving this article.
Yours sincerely,
Dr. Luca Éva Uhljar
Prof. Rita Ambrus
- 07. 2022. Szeged, Hungary

Reviewer 2 Report
The authors have done well improving the manuscript, however, they should address the below comment appropriately.
- It is unusual that the 10% PVP showed bead. Something is wrong here. By checking the study parameters, it seems that the parameters are right that’s why it is weird not to have proper fibers at the concentration especially by using the PVP MW. Please clarify this and compare the interpretation to previous studies using similar PVP MW.
- If the 10% PVP was too viscous, why didn’t the authors choose 7 or 8% instead?
- Why the conc of Cipro in the cytotoxicity study is different from the formulations?
- The author should have considered the microbiology assay (zone of inhibition) better than the cytotoxicity. Does the antibiotic retain its efficiency? The provided response wasn’t justifiable
- Discussion is weak and needs improvement in almost all result sections. Compare the results you obtained with previous studies. Interpret the obtained results well, as interpretation lacks in some of the result sections
Author Response
The authors have done well improving the manuscript, however, they should address the below comment appropriately.
Thank you again for your questions. We are trying our best to answer your questions and clarify the ideas.
It is unusual that the 10% PVP showed bead. Something is wrong here. By checking the study parameters, it seems that the parameters are right that’s why it is weird not to have proper fibers at the concentration especially by using the PVP MW. Please clarify this and compare the interpretation to previous studies using similar PVP MW.
If the 10% PVP was too viscous, why didn’t the authors choose 7 or 8% instead?
Thank you for your questions.
Beading in electrospinning happens when the electrospinning solution is not viscous enough and the jet from the Taylor cone is not continuous.
In our lab, we use PVP (mainly the high, 1.3M molecular weight) and we consistently see that in order to make nanofibres without beads, we need at least 12 wt% PVP concentration. This depends on the solvent, and temperature, as viscosity depends on these factors as well.
[1,2]
Apparently, for the used system, 10 wt% was not viscous enough, that’s why we didn’t use lower concentrations. However, as we mentioned before, in our previous experiments we used the same PVP MW but with lower CIP content. In that case, proper fibers could be produced [3].
- Nartetamrongsutt, K.; Chase, G.G. The Influence of Salt and Solvent Concentrations on Electrospun Polyvinylpyrrolidone Fiber Diameters and Bead Formation. Polymer 2013, 54, 2166–2173, doi:10.1016/j.polymer.2013.02.028.
- Munir, M.M.; Suryamas, A.B.; Iskandar, F.; Okuyama, K. Scaling Law on Particle-to-Fiber Formation during Electrospinning. Polymer 2009, 50, 4935–4943, doi:10.1016/j.polymer.2009.08.011.
- Uhljar, L.É.; Kan, S.Y.; Radacsi, N.; Koutsos, V.; Szabó-Révész, P.; Ambrus, R. In Vitro Drug Release, Permeability, and Structural Test of Ciprofloxacin-Loaded Nanofibers. Pharmaceutics 2021, 13, 556, doi:10.3390/pharmaceutics13040556.
Why the conc of Cipro in the cytotoxicity study is different from the formulations?
Thank you for your question again. We are trying to give a clearer answer now. The cytotoxicity test was performed as follows. The most concentrated solution of CIP was prepared. Due to the poor water solubility of CIP, this was 163 µg/mL. Then the different nanofibrous samples were dissolved in the medium. In the solution preparation, it was taken into account that the nanofibers contained 5 and 10 wt% of CIP. Since we worked with very small sample volumes, the mass measurement of the nanofibres was difficult. We could only achieve approximately the same concentrations. However, it can be seen that the difference between 163 and 157 and 81 and 78.5 for concentrations measured in µg/mL is so small that these quantities are comparable.
Furthermore, we used serial dilution in the test, which means that the concentration in each case was half of the previous concentration as diluted in the row. This meant that if there was a small difference in the initial concentration, this caused an increasing variation in each value as the series progressed.
The author should have considered the microbiology assay (zone of inhibition) better than the cytotoxicity. Does the antibiotic retain its efficiency? The provided response wasn’t justifiable.
Thank you for your suggestion. For antibiotics, microbiological testing may indeed be important. In the future, we plan to further develop this formulation into a per os or/and topical formulation that can be tested in microbiological and in vivo studies. Discussions are currently ongoing with the cooperation partners, but unfortunately, we have not had the opportunity to carry out this test. In the future we are going to fulfill this.
Discussion is weak and needs improvement in almost all result sections. Compare the results you obtained with previous studies. Interpret the obtained results well, as interpretation lacks in some of the result sections.
Thank you for your comment. We have improved the Conclusion section and compared the results with other papers’. The new parts can be seen with blue color.
- Conclusions
Nanofibers as amorphous solid dispersions can be formulated as orodisperse films or oral medicines while with the suitable polymer, fast disintegration and rapid release can be achieved. This paper presented the production and investigation of CIP-loaded PVP nanofibers prepared by nozzle-based and nozzle-free ES methods. Nanofibers with 5% and 10% CIP concentration were fabricated by the conventional single-nozzle ES. By the nozzle-free method, 5% CIP-loaded nanofibrous samples were produced from the same electrospinning solutions. Comparing the nanofibers, we found that the preparation method had no influence on the drug carriers as amorphous solid dispersions because both methods could amorphize the CIP. The amorphising effects of ES has been confirmed in several previous studies [29,44,49,54,55,62]. Also, the presence or absence of the nozzle had no effect on the in vitro drug release, as the dissolution of the CIP was complete and fast from the nanofibers and occurred according to Korsmeyer-Peppas model or first-order kinetics. This result was similar to the preliminary study in which CIP was present in lower concentrations in PVP nanofibers [29]. It also correlated well with the results of CIP-loaded PVP nanofibers investigated by other research group [63]. Finally, none of the prepared nanofibrous samples were cytotoxic according to the MTT test on CaCo-2 cell lines. This study may suggest further processing steps, which might include the production of an orodisperse film by pressing or production of per os drugs by filling the fibers into gelatin capsules [64,65].
However, it can be concluded that nozzle-free ES has several advantages. These are, based on the studies performed, a more homogeneous distribution of the active ingredient within the nanofiber mat, a higher encapsulation efficiency, and a longer stability. An additional advantage may be the increased productivity due to the fact that ES from the free solution surface allows the formation of several Taylor cones at the same time. On the other hand, this also leads to the disadvantage of the nozzle-free method, which is the wide variability in fiber diameters.
The study also focused on the stability of nanofibers. In this context, SEM, XRPD and in vitro release studies were carried out, which led to two types of conclusions. The first conclusion was that aged morphology, fusion of the fibers or partial recrystallisation of the CIP did not affect the in vitro drug release. Based on these results, it is important to check for drug release in all stability studies of nanofibers. As a second conclusion, the stability results suggested that NF-5 was sufficiently stable for 16 months in terms of morphology and 26 months in terms of amorphousness and in vitro release. Similarly, the amorphous form of API was preserved in nanofibers over 12 month according other studies [58,60]. In contrast, the morphology of the SN-5 sample remained unchanged up to 26 months, but the CIP started to recrystallize at 8 months and its release slowed down significantly at 16 months. Thus, the nanofibers prepared with nozzle-free ES was found to be more stable during the 26-month storage in a desiccator.
Overall, nanofiber production by nozzle-free ES equipment can not only provide a good alternative to the nozzle-based ES but also produces nanofibers with improved properties. Thus, it could be worthwhile to develop the method further and use the resulting nanofibers as drug delivery systems.
- Ajmal, G.; Bonde, G.V.; Thokala, S.; Mittal, P.; Khan, G.; Singh, J.; Pandey, V.K.; Mishra, B. Ciprofloxacin HCl and Quercetin Functionalized Electrospun Nanofiber Membrane: Fabrication and Its Evaluation in Full Thickness Wound Healing. Artificial Cells, Nanomedicine, and Biotechnology 2019, 47, 228–240, doi:10.1080/21691401.2018.1548475.
- Contardi, M.; Heredia-Guerrero, J.A.; Perotto, G.; Valentini, P.; Pompa, P.P.; Spanò, R.; Goldoni, L.; Bertorelli, R.; Athanassiou, A.; Bayer, I.S. Transparent Ciprofloxacin-Povidone Antibiotic Films and Nanofiber Mats as Potential Skin and Wound Care Dressings. European Journal of Pharmaceutical Sciences 2017, 104, 133–144, doi:10.1016/j.ejps.2017.03.044.
- Panda, B.P.; Wei, M.X.; Shivashekaregowda, N.K.H.; Patnaik, S. Design, Fabrication and Characterization of PVA/PLGA Electrospun Nanofibers Carriers for Improvement of Drug Delivery of Gliclazide in Type-2 Diabetes. In Proceedings of the The 1st International Electronic Conference on Pharmaceutics; MDPI, December 1 2020; p. 14.
- Partheniadis, I.; Athanasiou, K.; Laidmäe, I.; Heinämäki, J.; Nikolakakis, I. Physicomechanical Characterization and Tablet Compression of Theophylline Nanofibrous Mats Prepared by Conventional and Ultrasound Enhanced Electrospinning. International Journal of Pharmaceutics 2022, 616, 121558, doi:10.1016/j.ijpharm.2022.121558.
Thank you again for your time and help improving the content of the article.
Yours sincerely,
Dr. Luca Éva Uhljar
Prof. Rita Ambrus
- 07. 2022. Szeged, Hungary

Round 3
Reviewer 2 Report
I have asked the authors to improve the discussion and not the conclusion. It is preferable to cite statements in conclusion, especially since you are not talking about introducing a new thing from a future perspective.
"Discussion is weak and needs improvement in almost all result sections. Compare the results you obtained with previous studies. Interpret the obtained results well, as interpretation is lacking in some of the result sections."
Author Response
I have asked the authors to improve the discussion and not the conclusion. It is preferable to cite statements in conclusion, especially since you are not talking about introducing a new thing from a future perspective.
Thank you. There was a misunderstanding. The sentences inserted in the conclusion are meant to be the parts of the discussion.
These citations had been made in the conclusion, also, four new references (62-65) had been provided.
"Discussion is weak and needs improvement in almost all result sections. Compare the results you obtained with previous studies. Interpret the obtained results well, as interpretation is lacking in some of the result sections."
The sentences were moved to the appropriate part of the Result and Discussion and further additions were made. The changes are marked with turquoise color.
Thank you again, we believe that this time you would be completely satisfied with our work.
